# Mitochondria remodeling during endometrial stromal cell decidualization

Marco Dalla Torre[1], Daniele Pittari[1], Alessandra Boletta[2], Laura Cassina[2], Roberto Sitia[1,2], Tiziana Anelli[2]

**Upon hormonal stimulation, uterine endometrial stromal cells undergo a dramatic morpho-functional metamorphosis that allows them to secrete large amounts of matrix proteins, cytokines, and growth factors. This step, known as decidualization, is crucial for embryo implantation. We previously demonstrated how the secretory pathway is remodelled during this process. Here we show that hormonal stimulation rapidly induces the expression of many mitochondrial genes, encoded in both the mitochondrial and the nuclear genomes. Altogether, the mitochondrial network quadruples its size and establishes more contacts with the ER. This new organization results in the increased respiratory capacity of decidualized cells. These findings reveal how achieving an efficient secretory phenotype requires a radical metabolic rewiring.**

## Introduction

Endometrial stromal cells (EnSCs) play fundamental roles in endometrial tissue homeostasis, control of embryo implantation, and the development of the maternal components of the placenta (Gellersen & Brosens, 2014). In the first phase of the menstrual cycle, EnSCs undergo rapid proliferation. At this stage, they are similar to conventional fibroblasts in terms of morphology and secretion activity. However, when progesterone levels rise in the post-ovulatory phase, EnSCs stop dividing and start a complex differentiation process called *decidualization*, acquiring an epithelioid, highly secretory phenotype (Dunn et al, 2003; Anelli et al, 2022). EnSC differentiation is part of a synchronized rearrangement of the components of the endometrial tissues (luminal and glandular epithelia, vasculature, residing immune cells, and extracellular matrix) that is fundamental for successful implantation of the embryo (Dunn et al, 2003; Gellersen & Brosens, 2014). Decidualized EnSCs are pivots in endometrium reorganization, mostly through the wide array of proteins they produce and secrete: they reshape the surrounding tissue organization, releasing

different types of collagen (Carbone et al, 2006; Shi et al, 2020) and other extracellular matrix proteins (Okada et al, 2014; Schatz et al, 2016), matrix metalloproteases and protease inhibitors (Dunn et al, 2003; Sharma et al, 2016). In addition, decidualized EnSCs coordinate the activity of other cell types through the release of cytokines, chemokines (Petracco et al, 2012; Vinketova et al, 2016), growth factors (Albrecht & Pepe, 2010; Okada et al, 2018), and pro- and anti-angiogenic molecules (Dunn et al, 2003; Albrecht & Pepe, 2010).

To cope with their new tasks, decidualizing EnSCs stop proliferating and reshape their architecture. However, differently from what observed in other professional secretory cell types (van Anken et al, 2003; Shaffer et al, 2004), EnSCs increase their secretory output with a limited expansion of the ER (Anelli et al, 2022). Instead, the Golgi complex enlarges, driven by the activation of CREB3L1 and CREB3L2 (Kajihara et al, 2014; Anelli et al, 2022; Pittari et al, 2022).

Despite its crucial importance in human reproduction, much remains to be understood about the processes of decidualization. How do decidualizing EnSCs adapt their energy metabolism to the novel demands of their secretory phenotype? Indeed, the higher activity in the exocytic pathway, and the other tasks that decidualization entails, require large amounts of chemical energy. Formation of a peptide bond requires on average 5 ATP molecules (Princiotta et al, 2003), whereas the addition of a single monosaccharide to a glycoprotein requires 2 ATP molecules (Aebi, 2013; Freeze et al, 2022). Chaperone-mediated quality control, proteasomal degradation, and vesicular transport also require abundant energy (Martínez et al, 2020; Wiseman et al, 2022). To learn more about the energetics of decidualization, here we investigate the dynamic rearrangements of the mitochondrial network occurring in EnSCs in response to hormonal stimulation.

## Results

### Mitochondrial network enlarges and reorganizes in decidualization

Upon in vitro treatment with progesterone (MPA) and cAMP, TERT-immortalized human EnSCs (T-HESC, Krikun et al, 2004) undergo in vitro decidualization (Brosens et al, 1999; Anelli et al, 2022; Pittari

---

[1]Università Vita-Salute San Raffaele, Milan, Italy    [2]IRCCS Ospedale San Raffaele, Division of Genetics and Cell Biology, Milan, Italy

Correspondence: anelli.tiziana@hsr.it

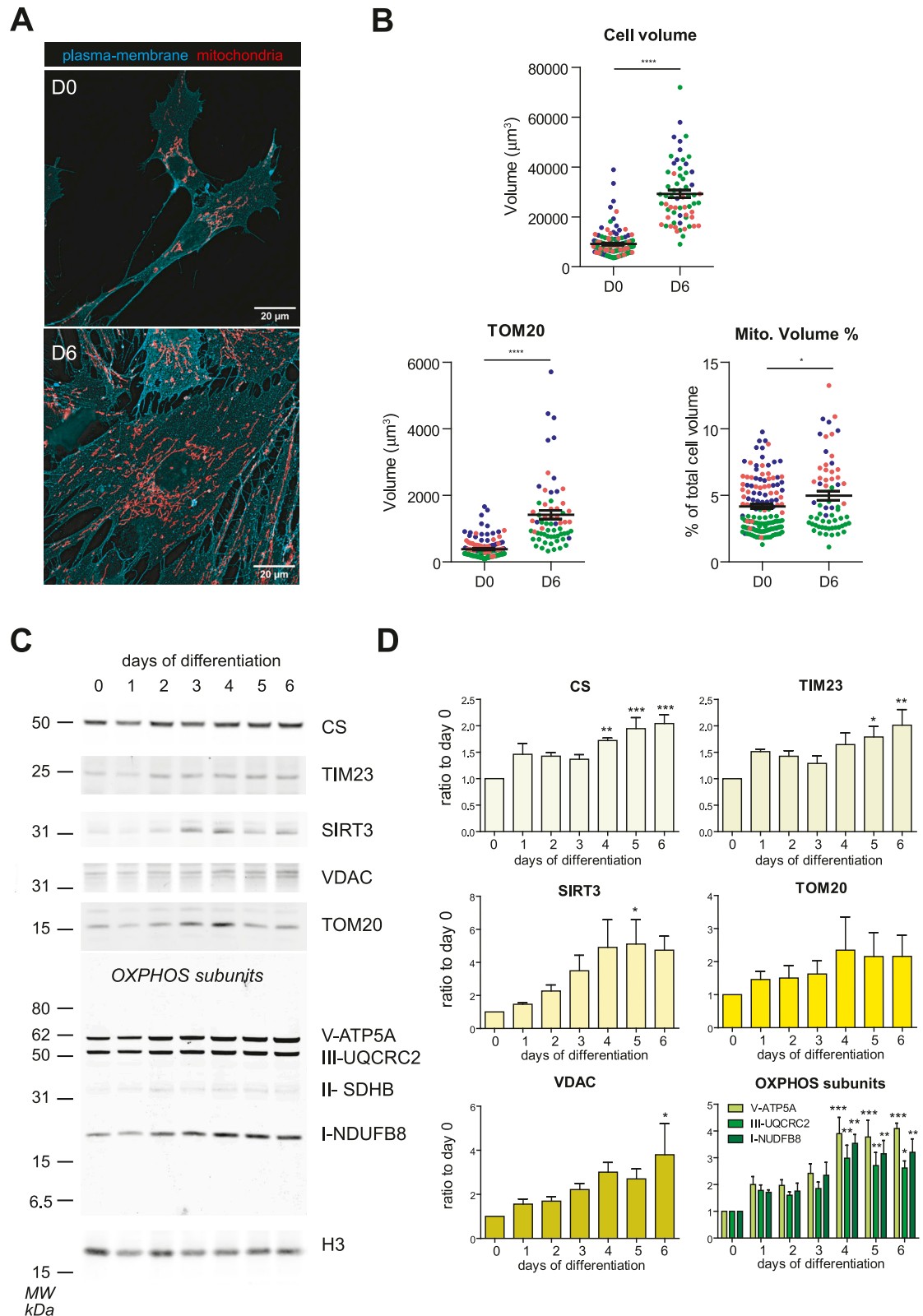

**Figure 1. Decidualization entails mitochondria network enlargement.**
**(A)** Mitochondria reorganize during decidualization. Proliferative and decidualized cells were fixed and immunostained for HLA-1, to visualize the plasma membrane (cyan), and for the mitochondrial marker TOM20 (red). Note the enlargement of the cell volume and the clear expansion of the mitochondrial network in decidualized EnSCs. 100× magnification, bar: 20 μm. **(B)** Mitochondrial volume significantly increases upon decidualization. Immunofluorescence-based morphometric analysis of total

et al, 2022), up-regulating known decidualization markers as pre-viously shown (Brosens et al, 1999; Anelli et al, 2022; Pittari et al, 2022 and Fig S1A), and abandoning their elongated, fibroblast-like shape to adopt a much larger, epithelioid morphology (Fig 1A). Staining with antibodies specific for Translocase of the Outer Membrane 20 (TOM20, red signal), revealed an extensive reorganization of mi-tochondria in differentiated cells (D6) with respect to untreated controls (D0). The size of the mitochondrial network increases dramatically and changes its organization, acquiring a radial dis-tribution towards the cell periphery (Fig 1A).

To better characterize the changes in the mitochondrial network in relation to the variations of the total cell volume (Anelli et al, 2022, Fig 1B, upper panel), we performed morphometric analyses, using an experimental approach previously described (Anelli et al, 2022; Pittari et al, 2022). Upon complete decidualization, corre-sponding to 6 d of culture with MPA and cAMP, mitochondria in-creased their absolute volume by 4.5-folds on a per cell basis (Fig 1B, bottom left panel). Importantly, the mitochondrial network also increased relative to total cell volume (Fig 1B, bottom right panel). To provide a comparison, the ER/total cell volume ratio decreases slightly under the same conditions, although the organelle size increased in absolute terms (Anelli et al, 2022). Surprisingly, therefore, decidualization—a secretory differentiation program—does not require a massive enlargement of the ER, the entry site for the secretory pathway (Anelli et al, 2022), but instead entails a dramatic enlargement of the mitochondrial compartment.

In accordance with the enlargement of the mitochondrial net-work, Western blotting analyses confirmed the strong increase, at the protein level, of different mitochondrial markers. Fig 1C shows a representative Western blot analysis, the quantification of three or more of which is reported in Fig 1D. Clearly, the levels of citrate synthase (CS) and sirtuin 3 (localized in the mitochondrial matrix), Translocase of the Inner Membrane 23 (a component of the inner mitochondrial membrane), Voltage Dependent Anion Channel (VDAC localized on the outer mitochondrial membrane) and OXPHOS subunits increase during decidualization.

Three-dimensional reconstructions of the immunofluorescence images (Fig 2A) revealed the dramatic changes in the mitochondrial network morphology upon decidualization (D0 versus D6). Con-sidering the length along the main axis of individual mitochondria, decidualized cells contain a higher proportion of tubular, elongated mitochondria (Fig 2B). Mitochondria between 5 and 10 $\mu$m of length double in frequency during differentiation, whereas mitochondria longer than 10 $\mu$m triplicate in proportion (Fig 2C). These figures may be under-evaluating the number of very long mitochondria in decidualized cells because their perinuclear concentration makes segmentation and identification of individual organelles hard to perform. Both proliferative and decidualized cells also contain numerous small mitochondria (1–2 $\mu$m in length), ruling out general

defects in the network organization, with dysfunctional mito-chondria adopting a hyperfused state (Westermann, 2010). This was further confirmed by live imaging with the fluorescent probe Mitotracker Red. As expected, we observed a core of elongated and static mitochondria, and several smaller mitochondria, actively transported along the cell, and both fission and fusion events (Fig S1B).

### Mitochondria-ER contacts increase during decidualization

Next, we exploited transmission electron microscopy (TEM) to characterize the morphological changes in the mitochondrial network at an ultrastructural level. Coherently with what was ob-served in immunofluorescence (Figs 1A and 2A–C), TEM images of decidualized cells showed numerous highly elongated mitochon-dria, rarely detected in proliferative cells (Fig 2D). Their detection by TEM was remarkable, considering that (because of the thin sections used and the possibility of fragmentation artifacts during fixation) the technique is rarely representative of the physiological archi-tecture of the mitochondrial network (Bereiter-Hahn, 1990). Indeed, in both proliferative and decidualized cells, most mitochondria appeared as "bean-like" objects that, when organized in "chains," likely correspond to long, individual mitochondria crossing the cutting plane such as a snake (as indicated by yellow arrows in Fig S2A). The accumulation of elongated mitochondria in decidualized cells is confirmed by additional morphological parameters, such as roundness and aspect ratio (respectively, decreasing and in-creasing with decidualization, see Fig S2B and C and see the Ma-terials and Methods section). The presence of normal cristae makes non-physiological hyperfusion an unlikely cause for mitochondrial elongation (Kondadi et al, 2019). Intriguingly, proliferative cells contained slightly more mitochondria with morphological alter-ations than differentiated cells (red arrows in Fig S2A), possibly reflecting different metabolic conditions and/or stringent quality control of the organelles in cells at D6 of differentiation. As decidualized cells are expected to be long-lived upon embryo implantation, prompt removal of damaged mitochondria may represent a paramount measure (Onishi et al, 2021; Kodali et al, 2022). Indeed, autophagy seems to be crucial for proper decidu-alization (Mestre Citrinovitz et al, 2019).

TEM images revealed more and closer associations between smooth ER and mitochondria (Fig 2D), part of which we interpret as organized contact sites (mitochondria-ER contact sites, MERCs) rather than random proximity in crowded cells. These regions of tight association between the ER and the mitochondria are crucial for the exchange of calcium (Szabadkai et al, 2006), lipid biosyn-thesis intermediates (Vance, 2014), and reactive oxygen species (Debattisti et al, 2017), and mitochondrial fusion and fission events (Friedman et al, 2011; Abrisch et al, 2020). Focussing on associations

---

cell volume (HLA-1) and of the mitochondrial network (TOM20), in three independent experiments (see the Materials and Methods section for details). Results are expressed as absolute volumes and relative mitochondria abundance. The plots show mean ± SEM and individual values of each group, t test. **(C, D)** Decidualization increases the expression of mitochondrial proteins. Western blot assays (C) and densitometric quantification (D) of different mitochondrial proteins localized in diverse mitochondrial subcompartments. The protein extracts corresponding to the same number of cells (100,000) were loaded in each lane under reducing conditions (see the Materials and Methods section for details). MW markers are shown on the left. Images were quantified with Image J and normalized first on H3 for loading, and then on day 0 for each staining. The graphs show the mean (+/− SEM) of a minimum of three independent experiments. One-way ANOVA followed by Dunnett's multiple comparison test on day 0 was performed.

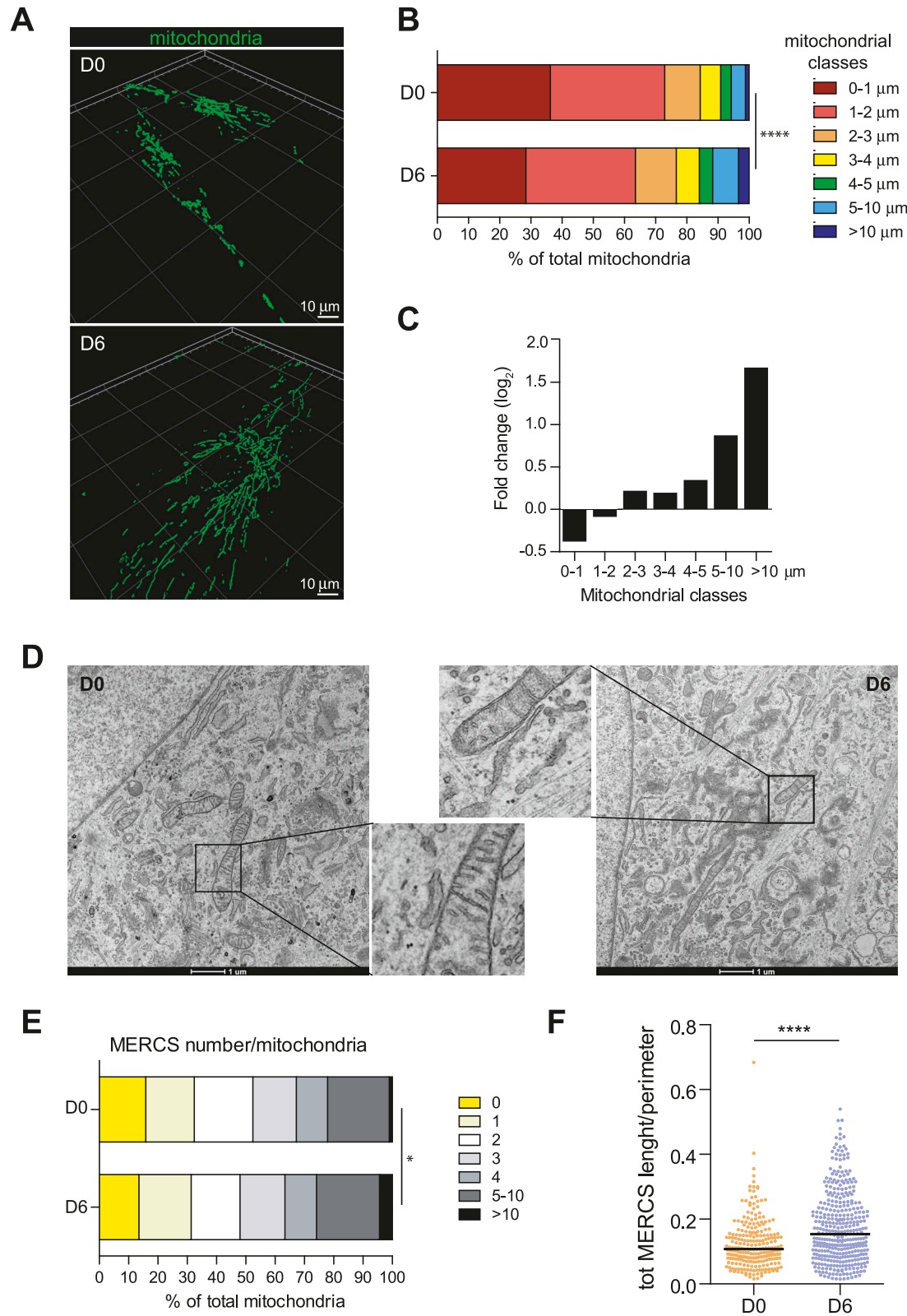

**Figure 2. Decidualization reshapes the mitochondrial network and its contacts with the ER.**
**(A)** Decidualization reshapes the mitochondrial network morphology. Three-dimensional representation of the mitochondrial network of proliferative and decidualized T-HESCs (days 0 and 6 of hormonal stimulation, respectively). The reconstruction was performed with Arivis 4D software based on representative immunofluorescences of TOM20 at 100× magnification. Scale bar: 10 μm. **(B, C)** Mitochondria elongate during decidualization. Morphometric analyses of mitochondrial elongation: analysis of the

of ≤30 nm between mitochondria and ER membranes devoid of ribosomes (Sorrentino et al, 2022), and excluding possible longer distance associations, the number and the length of the MERCs were significantly increased in decidualized cells (Fig 2E), especially after normalization to the size of the mitochondria involved (Fig 2F). As we didn't observe dramatic differences in the overall cellular crowding, we conclude that decidualization entails increased functional cooperation between mitochondria and ER.

In EnSCs, the rough ER, defined by the staining for markers involved in protein folding and quality control, such as calreticulin (CRT) or protein disulfide isomerase (PDI), concentrates in the perinuclear area (Anelli et al, 2022). In contrast, ER structural components such as reticulon 4 (RTN4), display a broader distribution, with many RTN4+PDI$^{neg}$ tubules being detectable also in the cell periphery (Fig S2D). Immunofluorescence analysis suggests that, especially in decidualized cells, mitochondria localize more closely with RTN4+ than with PDI+ ER, possibly reflecting different functional roles of the MERCs involved. Providing energy to ER protein folding is indeed a key function of ER-mitochondria interactions, but it does not necessarily require close and organized contacts. The interactions between mitochondria and the RTN4+$^{-}$PDI$^{neg}$ smooth ER tubules could instead sustain the exchange of lipid biosynthesis intermediates, which is fundamental for the expansion of the cellular membranes (see also Figs 1A and 2A and S3), and for the exchange of calcium and/or reactive oxygen species in well-organized contact sites (Rowland & Voeltz, 2012).

### Increased mitochondrial respiration in decidualized cells

As determined by the Seahorse Mito Stress Test assay (Gu et al, 2021), decidualization significantly modifies the mitochondrial respiratory profile (Fig 3A): decidualized EnSCs increase both basal and maximal respiration and the spare respiratory capacity on a *per cell* basis. Enhanced mitochondrial metabolic activity (Fig 3A and B) is likely important to cope with the novel metabolic needs. Accordingly, increased basal oxygen consumption is paralleled by a higher efficiency in the process. Indeed, the amount of oxygen devoted to ATP production is higher in decidualized cells (Fig 3B).

The rewiring of the energy metabolism in decidualized cells entails a contemporary decrease in basal extracellular acidification rate (ECAR), which estimates glycolytic activity (Fig S3), thus pointing to a shift in the basal energetic metabolism, from a glycolytic to a more aerobic phenotype (as shown in the graph in Fig 3C).

### Transcriptional control of mitochondrial remodeling

Unbiased transcriptomics and over-representation analysis of the up-regulated genes (Fig 4A and B) (Pittari et al, 2022) confirmed a prompt induction of the OXPHOS assembly factors during decidualization, followed by the accumulation of the respiratory complex subunits (Fig 4A and B). The dramatic transcriptional increase in nuclear-encoded OXPHOS subunits was also confirmed by qRT-PCR (Fig 4C). Interestingly, mitochondrial-encoded subunits are down-regulated initially but start to accumulate 36 h after hormonal stimulation, possibly underlying a more complex regulation pattern (Fig 4B).

Although lacking the depth of a complete functional characterization, this approach provides a general overview of the changes in mitochondrial activities along decidualization. A wider analysis is provided as Supplementary Material (Tables S1, S2, and S3). Genes related to ROS detoxification and lipid biosynthesis are also induced (Figs 4A and B and S4A), which comes in support of the important role of MERCs in decidualized cells. Genes involved in mitochondrial calcium import and homeostasis—another function that depends on MERCs (Patergnani et al, 2011)—are up-regulated in the second phase of the differentiation program (Figs 4B and S4B), except for the negative regulator of the Mitochondrial Calcium Uniporter (Raffaello et al, 2013; Sancack et al, 2013), *MCUB*, which is instead dramatically down-regulated. This would suggest an increased mitochondrial capacity to rapidly uptake calcium (D'Angelo & Rizzuto, 2023).

As to the Krebs cycle enzymes, they do not figure among the most down-regulated or up-regulated pathways (see Tables S1, S2, and S3). qRT-PCR analyses (Fig S5A) indicated that both citrate synthase (*CS*) and alpha-ketoglutarate dehydrogenase (*OGDH*) undergo a slight relative decline at the beginning of decidualization returning to the original levels at later stages. Conversely, glutamate dehydrogenase 1 (*GLUD1*), a key enzyme in mitochondria glutamine metabolism, and genes related to fatty acid beta-oxidation—acyl-CoA synthetase short-chain family member 1 (*ACSS1*) and carnitine palmitoyltransferase 1A and 1C (*CPT1A* and *CPT1C*)—are highly up-regulated (Fig S5B and C). These changes suggest a possible change in energy sources during the differentiation program.

Finally, genes related to mitochondrial DNA replication, maintenance, and repair are instead down-regulated (Fig 4A and B), probably reflecting a switch from proliferation (in which mitochondria are duplicated at every cell replication) to terminal decidualization. Taken together, the above data suggest that decidualization entails drastic changes in energy handling.

### Decidualization activates the master regulators of mitochondrial biogenesis

The induction of mitochondrial genes is a complex process, controlled by specific transcription factors. As most mitochondrial proteins are nuclear-encoded, their transcription mostly depends on the nuclear translocation of factors such as nuclear respiratory

---

main axis. **(B)** Mitochondria were grouped in arbitrary length classes, based on their main axes. Results were expressed in proportion to the total number of mitochondria analyzed in two independent replicates (≥25 cells for each condition). Kolmogorov-Smirnov test. **(C)** Fold change of the relative frequency of mitochondria belonging to different length classes. The results are expressed as log$_2$. **(D)** Transmission electron microscopy reveals close interactions between ER and mitochondria. Transmission electron microscopy of proliferative (D0, left) and decidualized (D6, right) T-HESCs. Details including contacts between ER and mitochondria are shown for both panels with the same magnification factor (3.5x). Representative images of three independent replicates. Scale bar: 1 $\mu$m. **(E, F)** ER and mitochondria increase their association during decidualization. TEM-based analyses of MERCs. **(E)** Classification of mitochondria based on the number of contact sites established with the ER. Results are expressed in proportion with the total number of mitochondria analyzed for each condition, *t* test. **(F)** Analysis of the total length of the MERCs established by each mitochondrion, normalized over the size of the mitochondrion itself (i.e., the perimeter of the object in TEM images), *t* test.

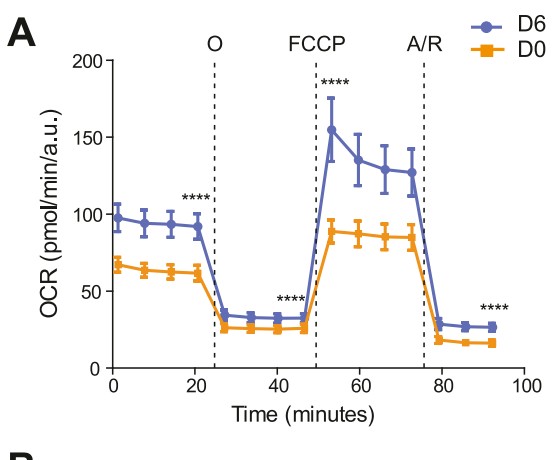

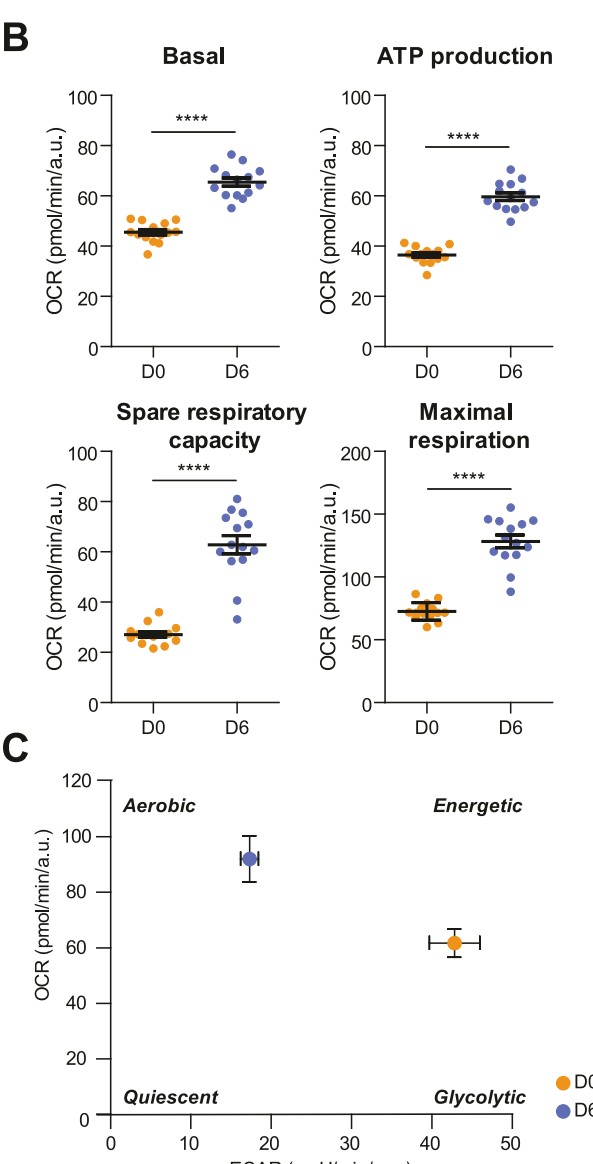

factor 1 (NRF1), nuclear respiratory factor 2 (NRF2), peroxisome proliferator-activated receptor alpha and gamma (PPARα, PPARγ) and PPARG coactivator 1 alpha (PGC-1α), with the latter playing a pivotal role for its capacity to interact with and regulate the activity of the others (Abu Shelbayeh et al, 2023). Immunofluorescence analyses confirmed that the translocation of PGC-1α into the nucleus is an early event in decidualization (Fig 5A). Its nuclear accumulation peaked 6 h after hormonal stimulation, remaining high also at 24 h. Not only is PGC-1α early translocated in the nucleus, but it is also highly transcriptionally induced, starting from day 2 of decidualization (Fig 5B). No significant changes in *NRF1* mRNA levels occur, whereas *NRF2* is induced at later stages (Fig 5B).

## Discussion

Almost 200 yr after the first medical descriptions of decidua in 1826 (Dewees, 1826), we sought to learn more about the molecular mechanisms underlying the spectacular metamorphosis of decidualizing EnSCs, and their cross-talks with other endometrial cell types. Our previously published results indicate that decidualizing EnSCs achieve a high secretory rate mainly by improving the efficiency—rather than the size—of their protein factory: indeed, with the exception of the Golgi apparatus, the relative abundance of exocytic organelles does not increase as in other secretory cell types (van Anken et al, 2003; Kirk et al, 2010; Anelli et al, 2022; Pittari et al, 2022).

Increased efficiency demands energy and better organization, which we surmise is provided by the massive structural and functional reshaping of the mitochondrial network that occurs during decidualization. Accordingly, functional analyses of decidualized EnSCs reveal a strong increase in mitochondrial respiration, highlighted by the increase in oxygen consumption. At the same time, aerobic glycolysis is reduced. Networks of elongated, highly interconnected mitochondria similar to those forming during decidualization are often observed in cells with high metabolic activity (Westermann, 2010; Gomes et al, 2011). These networks are more efficient than small fragmented mitochondria in terms of proton potential generation, energy distribution, and ROS scavenging, especially in peripheral areas of the cell (Skulachev, 2001; Napolitano et al, 2021).

A previous work (Ryu et al, 2020) described lower respiration and ATP production in decidualized cells. These results were obtained with a different experimental model (primary human EnSCs derived from myomas, immortalized and in vitro decidualized), whereas here we are using a cell line, likely with reduced variability in the differentiation program. The use of different experimental models could account for the discrepancy in the results. Moreover, we are aware that in our in vitro models, key parameters such as glucose concentration and oxygen tension might differ from those

**Figure 3. Decidualization entails a switch in EnSCs' energy metabolism.**
**(A)** Analysis of OCR measurement of one representative experiment in proliferative (D0) and decidualized (D6) T-HESCs, in basal condition and after sequential addition of oligomycin (O), FCCP, and antimycin/rotenone (A/R). 14 technical replicates for each experimental condition were analyzed. **(A, B)** Quantification of basal respiration, ATP production, spare respiratory capacity, and maximal respiration in (A). Data in bar plots are mean ± SD, *t* test. **(A, C)** Cell energy phenotype plot showing mitochondrial respiration (oxygen consumption rate, OCR) versus glycolysis (extracellular acidification rate, ECAR) rates of proliferative (D0) and decidualized (D6) T-HESCs in (A).

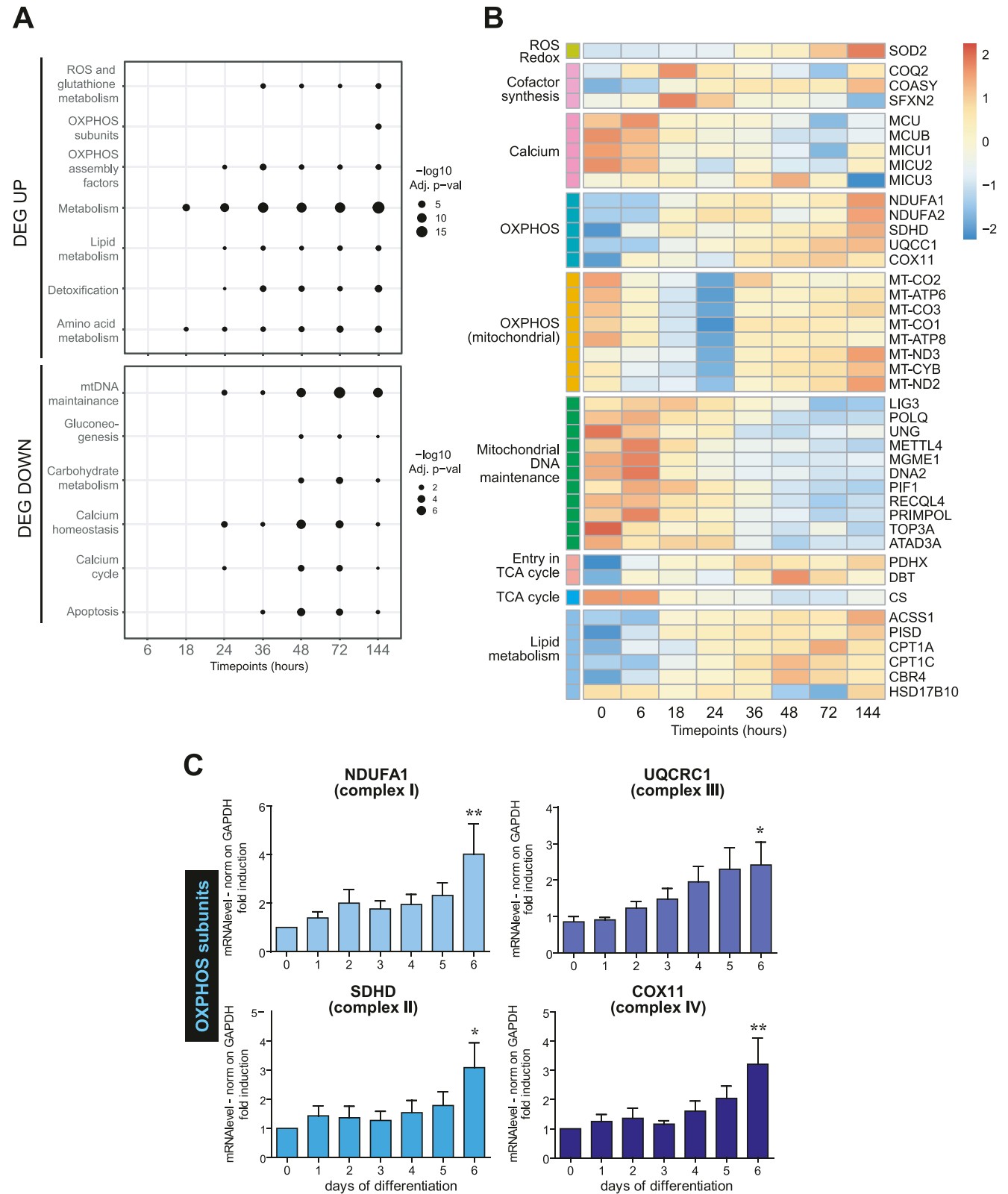

**Figure 4. Decidualization reshapes the transcriptional pattern of mitochondrial genes.**
**(A)** Time-course over-representation analysis for the genes up-regulated (DEG UP) or down-regulated (DEG DOWN) at different time points. MitoCarta terms associated with different biological functions of mitochondria were analyzed. Statistical significance of the over-representation analysis is graphically represented by the different sizes of the circles. **(B)** Heatmap of genes differentially expressed during the decidualization program. Normalized expression values were z-scaled and represented as a

experienced by EnSCs in vivo. Yet, the metabolic rewiring observed in our experiments is coherent with known aspects of physiological decidualization. Indeed, the development of spiral arteries provides the tissue with highly oxygenated blood (Pijnenborg et al, 2006), which would justify the early mitochondrial network enlargement and final EnSC switch to increased oxygen consumption. At the same time, decidualized stromal and epithelial cells accumulate glycogen granules, required to sustain the early stages of embryo development (Gellersen & Brosens, 2014). In this context, mitochondrial respiration probably represents the most efficient solution for the energy requirements of decidualized cells. Interestingly, decidualized EnSCs rewire their energetic metabolism differently from other examples of secretory differentiation. For instance, besides the massive development of the ER, B-to-plasma cell transition entails up-regulation of oxidative phosphorylation but also, transiently, glycolysis (Garcia-Manteiga et al, 2011).

Besides OXPHOS subunits, genes of mitochondrial proteins involved in lipid metabolism, and particularly enzymes for fatty acid beta-oxidation (*ACSS1*, *CPT1A*, *CPT1C*), are dramatically up-regulated during decidualization. This is true also for *GLUD1*, a key enzyme in mitochondria glutamine metabolism. This likely reflects a change in the prevailing energy sources in decidualized cells, which could allow cells to channel glucose in metabolic pathways different from energy production, including glycogen storage (Gellersen et al, 2014), glycoprotein biosynthesis (Aebi, 2013) or the pentose phosphate pathway (Alfarouk et al, 2020). Other functions of the mitochondrial network are also significantly modified by decidualization. Genes involved in the biosynthesis of phospholipids are up-regulated, coherently with the massive increase in membrane-bound organelles and transport vesicles. ROS detoxification (e.g., *SOD2*) is enhanced too. Both these fundamental functions depend on close interactions between the ER and mitochondria (Rowland & Voeltz, 2012). Accordingly, our imaging analyses revealed more and longer MERCs.

In a search for understanding the molecular mechanisms responsible for the mitochondrial rewiring observed during decidualization, we focused our attention on PGC-1α, the master regulator of mitochondrial biogenesis (Mastropasqua et al, 2018). As previously observed for the secretory pathway genes, the transcriptional reshaping of mitochondria begins immediately after the exposure of the cells to the decidualization stimuli, far before any increase in protein cargo synthesis (Anelli et al, 2022; Pittari et al, 2022). In line with this, we demonstrated that PGC-1α translocates to the nucleus immediately after hormonal stimulation. This result is particularly interesting, considering the role of this transcription factor also in the regulation of oxidative phosphorylation and lipid metabolism (Mastropasqua et al, 2018). Moreover, PGC-1α is highly expressed in decidualized cells (Bombail et al, 2008), and is directly involved in the transcriptional induction of PRL and IGFBP1 (Takagi et al, 2022). PCG-1α activation, therefore, can likely be considered

part of the differentiation program, independent from of the possible adaptation to massive protein production.

Thus, EnSCs prepare well in advance, activating a predetermined program that entails extensive organellar reshaping in preparation for their novel physiological tasks. In this way, they avoid proteotoxic stress, likely ensuring a longer life devoted to cuddling the growing embryo.

The detailed description of decidualization we provide here opens many questions. Considering the spatial distribution of ER subregions in EnSCs, for example, it would be nice to see whether MERCs play specialized functions in different areas of decidualizing cells. Our results are important from both a medical and a biological viewpoint. On the one hand, understanding mitochondrial dynamics in decidualization could help prevent recurrent implantation failure or endometriosis. On the other, learning more on different secretory differentiation programs could have broad implications in biotechnological drug production.

# Materials and Methods

## Cell culture

Experiments were performed on the TERT-immortalized human EnSC cell line T-HESC (ATCC CRL-4003; RRID:CVCL_C464). As from manufacturer's instructions, cells were maintained in 1:1 DMEM-Ham's F-12 mixture, with 3.1 g/l glucose and 1 mM sodium pyruvate, without phenol red, and supplemented with 1.5 g/l sodium bicarbonate, 1% ITS + Premix, 500 ng/ml puromycin, and 10% charcoal/dextran treated FBS. Medium was supplemented every 3–4 d with ascorbic acid (50 μg/ml). Decidualization was induced as previously described (Brosens et al, 1999; Gellersen & Brosens, 2014), by addition of 8-bromoadenosine 3′,5′ cyclic monophosphate (cAMP) 0.5 mM and medroxyprogesterone acetate (MPA) 1 μM to the culture medium; induction was repeated after 72 h, and day 6 (144 h) was considered as complete differentiation.

## Immunofluorescence

T-HESCs were plated on glass slides (0.13–0.16 mm thickness) and maintained in culture with or without decidualization induction. Cells were fixed in formalin solution (neutral buffered, 10%) for 10 min at RT and then washed with PBS.

Cells were permeabilized with 0.1% Triton X100 in PBS, and incubated with blocking solution (PBS-5% FCS) for 30 min at RT. Cells were then incubated with primary and secondary antibodies, diluted in blocking solution as indicated: PGC-1α polyclonal antibody (Invitrogen); anti-RTN4 A/B polyclonal antibody (Invitrogen); anti-

color gradient. Genes were selected and grouped based on the biological functions mediated by their products. Mitochondrial-encoded OXPHOS genes were considered a separate group based on the peculiarity of their expression mechanisms. **(C)** qRT-PCR analysis of genes involved in oxidative phosphorylation. The samples correspond to RNAs collected on different days of in vitro decidualization of T-HESC, with a minimum of six independent replicates. Results were normalized on *GAPDH* mRNA and represented as mean ± SEM. One-way ANOVA followed by Dunnett's multiple comparison tests on day 0 was performed.

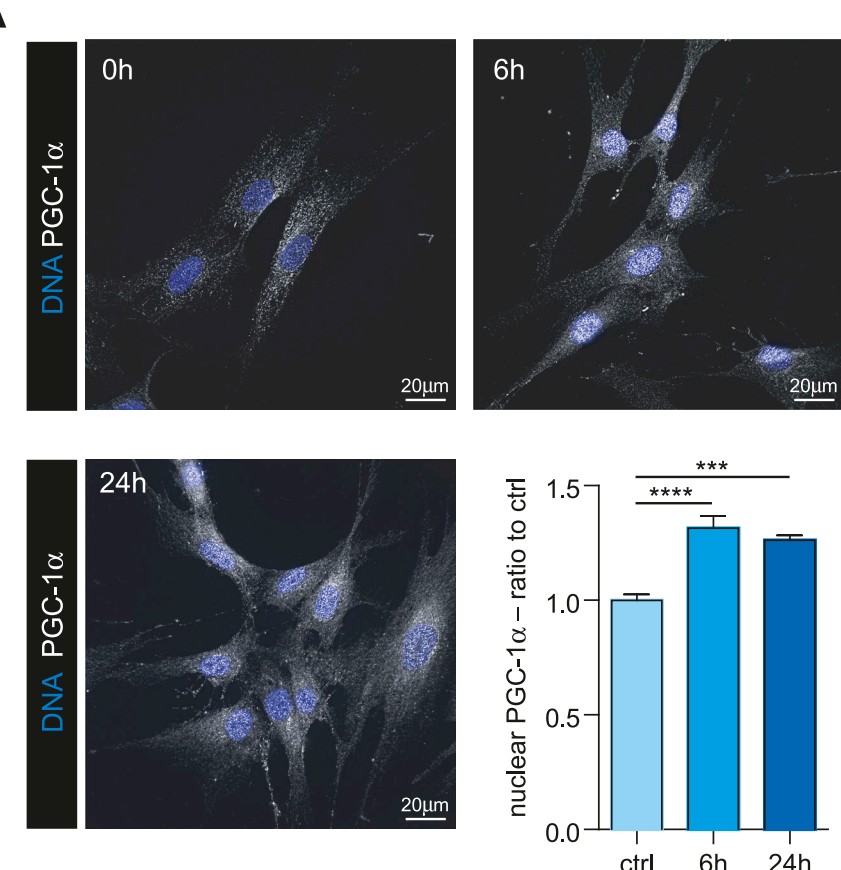

**Figure 5. Nuclear translocation of PGC-1α is an early event in T-HESC decidualization.**
**(A)** Immunofluorescence of PGC-1α at early time points of decidualization. Nuclei were counter-stained with DAPI to verify the nuclear translocation of the transcription factor. Images representative of two independent replicates. 60× magnification; scale bar: 20 μm. The graph at the bottom right represents the quantification of the nuclear translocation of PGC-1α: the bars show the average intensity of PGC-1α immunostaining in the nucleus, ± SEM (ctrl 42 cells, 6 h 66 cells, 72 h 50 cells). One-way ANOVA followed by Dunnett's multiple comparison tests on day 0 was performed. **(B)** qRT-PCR analysis of key transcription factors involved in mitochondria biogenesis. The samples correspond to RNAs collected on different days of in vitro decidualization of T-HESC, with a minimum of four independent replicates. Results were normalized on *GAPDH* mRNA and represented as mean ± SEM. One-way ANOVA followed by Dunnett's multiple comparison tests on day 0 was performed.

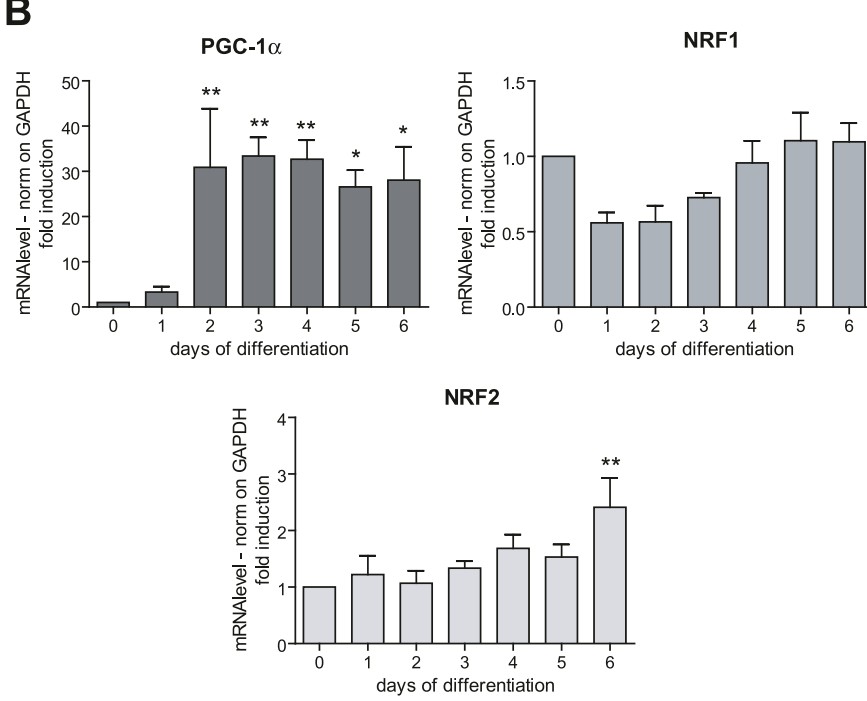

PDI monoclonal antibody (Invitrogen); Alexa Fluor secondary antibodies, 488, 546, or 647 (Thermo Fisher Scientific). Nuclei were counter-stained with DAPI, 1:5,000 in PBS.

For TOM20 staining, slides were incubated in blocking solution with 0.1% saponin for 30 min at RT, then with anti-TOM20 monoclonal antibody, 1:100 (BD Biosciences), and with Goat-anti-mouse

**Antibody list.**

| Target | Species | Dilution | Producer | Producer's code | RRID |
|---|---|---|---|---|---|
| Acetyl-Histone H3 | Rabbit | 1:1,000 (WB) | Upstate | 06-599 | AB_2115283 |
| Citrate synthase | Rabbit | 1:2,500 (WB) | Thermo Fisher Scientific | PA5-22126 | AB_11154268 |
| PGC-1α | Rabbit | 1:500 (IF) | Invitrogen | PA5-72948 | AB_2718802 |
| PDI | Mouse | 1:500 (IF) | Invitrogen | MA3-019 | AB_2163120 |
| RTN4 A/B | Rabbit | 1:250 (IF) | Invitrogen | PA1-41220 | AB_2254188 |
| Sirtuin 3 | Rabbit | 1:1,000 (WB) | Cell Signaling | D22A3 | AB_10828246 |
| Translocase of the Inner Membrane 23 | Rabbit | 1:1,000 (WB) | Abcam | ab116329 | AB_10903878 |
| TOM20 | Mouse | 1:1,000 (WB) 1:100 (IF) | BD Biosciences | 612278 | AB_399595 |
| Total OXPHOS Rodent WB Antibody Cocktail | Mouse | 1:2,000 (WB) | Abcam | ab110413 | AB_2629281 |
| VDAC | Rabbit | 1:500 (WB) | BioVision | 3594-100 | AB_2214790 |
| Mouse antibodies | Goat conj. Fluor 488 | 1:1,000 (WB) 1:500 (IF) | Thermo Fisher Scientific | A-11001 | AB_2534069 |
| Rabbit antibodies | Goat conj. Fluor 546 | 1:1,000 (WB) 1:500 (IF) | Thermo Fisher Scientific | A-11035 | AB_2534093 |
| Rabbit antibodies | Goat conj. Fluor 647 | 1:1,000 (WB) 1:500 (IF) | Thermo Fisher Scientific | A-21245 | AB_2535813 |
| Mouse antibodies | Goat conj. Fluor 647 | 1:1,000 (WB) 1:500 (IF) | Thermo Fisher Scientific | A-21235 | AB_2535804 |

secondary antibody, Alexa Fluor 647, 1:500 (Thermo Fisher Scientific); 0.1% saponin was added in both the antibody incubations and the washing steps. For membrane staining, slides were further incubated in PBS for 30 min at RT to remove saponin and then incubated again in blocking solution, for 30 min at RT. Cells were then incubated with anti-HLA mouse monoclonal antibody (W6-32) as hybridoma supernatant (RRID:CVCL_7872), and then with Goat-anti-mouse secondary antibody, Alexa Fluor 488, 1:500 (Thermo Fisher Scientific).

Images were acquired at 60× or 100× magnification with oil immersion objectives (Olympus 60X/1.42, 1-U2B933; Olympus 100X/1.45, 1-UXB240), with *DeltaVision Ultra microscope* (GE Healthcare), equipped with an sCMOS camera (pco.edge 4.2, PCO) at the Advanced Light and Electron Microscopy BioImaging Center (ALEMBIC) in Ospedale San Raffaele. Deconvolution was performed on the instrument workstation contextually with image acquisition, using the *Softworks* software (GE healthcare). Both original and deconvolved images were exported as 16-bit grayscale (in R3D and D3D format, respectively).

### MitoTracker staining and imaging

Mitochondrial staining was performed with MitoTracker Red CMXRos (Thermo Fisher Scientific), as from manufacturer's instructions. Briefly, MitoTracker was diluted in medium to a final concentration of 20 nM; cells were incubated with the dye for 30 min at 37°C, 5% $CO_2$, washed twice with PBS, and kept in complete medium until further processing.

Then, slides were either fixed in formalin solution and prepared for immunofluorescence, or transferred in a ring support for live imaging, covered with 500 $\mu$l Ringer buffer (140 mM NaCl, 2 mM $CaCl_2$, 1 mM $MgSO_4$, 1.5 mM $K_2HPO_4$, 10 mM glucose, 10 mM Hepes, buffered to pH 7.4) and imaged with *DeltaVision U. microscope* (GE

healthcare), 60× magnification (Olympus 60X/1.42, 1-U2B933) (ALEMBIC–OSR), in a 37°C, 5% $CO_2$ chamber and with automatic focus stabilization (UltimateFocus). To prevent mitochondrial fragmentation in response to thermal shock, all the buffers and mediums were pre-warmed to 37°C before use.

### Morphometry and morphological analysis

Raw image stacks (R3D format files) were deconvolved with Huygens Essentials software (CMLE, theoretical PSF, SNR 43.90, max iterations 94). Volume quantification was performed with Huygens Object Analyzer plugin: cells and mitochondria were identified with the threshold and seed segmentation approach, with parameters based on randomly selected images from different experiments. Intensity values of 100 and 1,500 were, respectively, chosen as threshold signals for cell membrane and mitochondria, representing the primary and secondary segmentation groups. The calculated volumes were converted into cubical microns (based on voxel size) and represented, both as absolute values and as the ratio between the mitochondrial and total volume of individual cells.

Morphological analysis was performed with the Arivis 4D software (version 3.6.2), using a custom analysis pipeline. For each object, the dimensions along the three axes and the sphericity parameters were measured. The objects were visually inspected, and the ones corresponding to an incorrect segmentation (visually recognized as not separated, or with a calculated main axis longer than 25 $\mu$m) were excluded from the analysis. Three-dimensional reconstruction of the mitochondrial network was performed on representative cells using the 3D visualization tool of Arivis, using the "glass" texture tool to improve volumetric rendering. All the described analysis pipelines are provided as Supplemental Data 1, Supplemental Data 2, and Supplemental Data 3.

**Primer list for qRT-PCR.**

| Target gene | Sense | Sequence |
|---|---|---|
| ACSS1 | forward | ATC GGA GCT GTC CAC ACA GT |
| ACSS1 | reverse | CGG AGT CCT TGG TTG AAG GT |
| COX11 | forward | TGT GCA TGC AAG TCT CCA GT |
| COX11 | reverse | TCT GTA AAA CGC CAG TGC AG |
| CPT1A | forward | GGATCCACGATTCCACTCTG |
| CPT1A | reverse | TACACGACGATGTGCTTGCT |
| CPT1C | forward | ACC CGG ACC CCA CAC TAC |
| CPT1C | reverse | TGG CAG TCG ACA TTT TCA GA |
| CS | forward | AGGAGCAGGCCAGAATTAAGA |
| CS | reverse | ACCAATCCCTTCATGCCTCT |
| GAPDH | forward | TGAAGGTCGGAGTCAACGGATTT |
| GAPDH | reverse | CATGTAAACCATGTAGTTGAGGT |
| GLUD1 | forward | TGATACCTATGCCAGCACCA |
| GLUD1 | reverse | TTGGCTGATGGGTTTACCAG |
| IGFBP1 | forward | TTACCTGCCAAACTGCAACA |
| IGFBP1 | reverse | CATTCCAAGGGTAGACGCAC |
| MCU | forward | CCTCCTCCTCCTTGATGACTTT |
| MCU | reverse | ATTCAGCGTTGCTGCATTTT |
| MCUB | forward | CAA CAG TTG GTT CAT TCC TTC A |
| MCUB | reverse | TCC ATC AAG GTA GAA GCT GAA ATC |
| MICU1 | forward | TTTGGAGCTGATCTGAAGGG |
| MICU1 | reverse | AATTCTCCCATCCACAGGGT |
| MICU2 | forward | TCT TCA GAT GCG TTT CTT TGG |
| MICU2 | reverse | GGA ATT CCA TTT CTT GAA TCT CTG T |
| MICU3 | forward | TGGAATTATGAAAGACAGACTCCA |
| MICU3 | reverse | CAGGCAGGATTTGAAAGTGG |
| NDUFA1 | forward | AGA TAG GCG CAT CTC TGG AGT |
| NDUFA1 | reverse | AGG GGT AGA TGG CCA TAA CTG |
| Nrf1 | forward | ACTCAGAACTGCCGCCTCTC |
| Nrf1 | reverse | GGTTTTCCCCGACCTGTAGA |
| Nrf2 | forward | CAATGAGGTTTCTTCGGCTACG |
| Nrf2 | reverse | AAGACTGGGCTCTCGATGTG |
| OGDH | forward | CTACACAACTCATGGCACCG |
| OGDH | reverse | ACGTCAGTGGGGTAGGGG |
| PGC1-a | forward | AAGCAAAGGGAGAGGCAGAG |
| PGC1-a | reverse | TGTCCGTGTTGTGTCAGGTC |
| PISD | forward | ATC TAA ATG GGC AAG GAC GC |
| PISD | reverse | GTT TGG TTC AGT CTC GGT GG |
| PRL | forward | AGCCAGGTTCATCCTGAAA |
| PRL | reverse | AGCAGAAAGGCGAGACTCTT |
| SDHD | forward | CCT ATC CCA GAA TGG TGT GG |
| SDHD | reverse | AGT GGA GAG ATG CAG CCT TG |
| SMDT | forward | TTGTTCCAGAGGATGATGATGA |

**Continued**

| Target gene | Sense | Sequence |
|---|---|---|
| SMDT | reverse | CTGCTGATAGGGAAGGCAGA |
| UQCRC1 | forward | TCC GAG CAG TCC TCT CAG C |
| UQCRC1 | reverse | CTC AGT CTC AAA ACG GCT GC |

### Transmission electron microscopy

Samples were fixed with 2.5% glutaraldehyde in 0.1 M sodium-cacodylate buffer pH 7.4, for 1 h at RT, washed three times with cacodylate buffer, and incubated with 1% osmium tetroxide and 1.5% potassium ferrocyanide in 0.1 M cacodylate for 1 h on ice. After repeated washes with distilled water, en bloc staining was performed overnight at 4°C with 0.5% uranyl acetate (in water).

Then, the samples were dehydrated in a graded ethanol series (30%, 50%, 70%, 80%, 90%, 96%, 5 min each; absolute ethanol, three washes, 10 min each), infiltrated in 1:1 ethanol/Epon 812 solution for 2 h, in 100% Epon twice for 1 h, and finally covered with a layer of Epon and polymerized in an oven at 60°C for 48 h.

A part of the sample was glued on an Epon block and sectioned with a Leica Ultracut UCT ultramicrotome. Ultrathin (70–90 nm) sections were collected on copper grids, stained with uranyl acetate and lead citrate, and finally examined with a transmission electron microscope (TALOS L120C; Thermo Fisher Scientific) at 120 kV.

Mitochondria were identified based on standard morphological parameters, that is, the presence of a double membrane and the identification of the cristae. In case of dubious identification, the objects were not included in the following analyses. The objects identified as mitochondria were visually inspected, to evaluate possible morphological alterations, either with a biological meaning or because of fixation artifacts. Mitochondria with (1) clear disruption of the outer membrane and/or of the cristae, or (2) swollen appearance, with a rounded shape, disruption of the cristae organization, and unusually large distance between the different membrane layers, were classified as "abnormal" in terms of morphology. No distinction was made in the classification was made between clear artifacts (e.g., damages because of the sectioning) or alterations with a biological meaning; the former were present only in a very limited number and appeared evenly distributed between the two conditions. Unless the alterations impaired the evaluation of ER-mitochondria contacts (e.g., significant loss of continuity of the mitochondrial membrane), the "abnormal" mitochondria were included in the subsequent analyses.

For each object, standard morphological parameters were calculated with ImageJ. In particular, surface and perimeter, aspect ratio (or A/R, i.e., the ratio between the major and the minor axes of the corresponding ellipse) and roundness (calculated as $4S/\pi M^2$, where S is the surface of the object and M is the major axis of the corresponding ellipse) were calculated for each object.

Morphometric analysis was performed as previously described (Sorrentino et al, 2022). A minimum of 50 images (D0: 83; D6: 79, corresponding to 378 and 473 mitochondria, respectively) were acquired in three independent experiments for each condition, and

all the mitochondria present in them were analyzed. The area and perimeter of the mitochondria were defined as regions of interest using the freehand selection tool of ImageJ. Then, each regions of interest was extended by 30 nm with the enlarge function, and all the ER structures present were scored as MAMs, measured, and compared with the total perimeter of the corresponding mitochondrion.

Regions of close proximity between ER and mitochondria, but in which other elements excluded the presence of a functional MERC (e.g., clear presence of ribosomes on the ER membrane, or of other structures between the organelles in the putative contact region) were excluded from the analysis. All the parameters acquired were analyzed and represented in GraphPad Prism, release 8.2.

### Seahorse assay for mitochondrial activity

Oxygen Consumption Rate (OCR) was measured using the Seahorse XF Mito Stress Test Kit (#103015-100; Agilent Technologies) on an XFe96 Analyzer (Agilent Technologies) following the manufacturer's instructions. Control and decidualized cells were plated in the 96-well Seahorse cell culture microplates 24 h before the experiment. Cell number was previously determined by pilot experiments: 10,000 and 20,000 cells for each condition were plated, with 15 replicated wells per experiment; 20,000 cells, corresponding to a confluence of 90%, were chosen as optimal. As from manufacturer's protocol, cells were sequentially treated with 1 µM oligomycin A (O), 1.5 µM FCCP, 0.5 µM (each) antimycin and rotenone (A/R). OCR and ECAR were measured for each well at the indicated time points. At the end of the experiment, cell numbers were normalized using CyQuant Cell Proliferation Assay (#C35011; Thermo Fisher Scientific). All the analyses were performed with the Agilent Seahorse Wave Desktop software (release 2.6, Agilent Technologies). Respiratory parameters were calculated as follows. Minimal Respiration: OCR after A/R injection; Basal Respiration: last measurement before O injection, minus Minimal Respiration; ATP Production: last measurement before O injection, minus the minimum OCR value after O injection; Maximal Respiration: maximal OCR after FCCP, minus Minimal Respiration; Spare Respiratory Capacity: Maximal Respiration, minus Basal Respiration. Two independent experiments were performed.

### Bulk RNASeq and data analysis

Cell samples were collected in duplicate before the stimulus (0 h) and at 6, 18, 24, 36, 48, 72, and 144 h after the start of the decidualization protocol. RNA was harvested using TRIzol (Invitrogen) according to the manufacturer's instructions. The library was prepared starting from 500 ng of total RNA per sample and processed using the TruSeq Stranded mRNA kit, according to the manufacturer's protocol. The sequencing step was carried out on an Illumina HiSeq 3000 platform. Sequencing data were analyzed using snakePipes with standard parameters (Bhardwaj et al, 2019). The pipeline uses STAR aligner to map the reads to the human genome (genome build hg38). FeatureCounts was used to count the number of reads per annotated gene using gencode gtf file (release 27).

Differential expression analysis (DEA), over-representation analysis, time series clustering and data visualization were performed using the R programming language (v. 4.1.0). Count matrix was pre-filtered before DEA by removing all genes showing a total number of counts, across all the samples, less than 30. DEA design consists of a pairwise comparison of each timepoint replicates versus untreated ones using the DESEQ2 package (v. 1.32.0) (Love et al, 2014). DEGs were called based on adjusted $P$-value < 0.01 and absolute $\log_2$ fold change >0.5. Over-representation analysis for MitoCarta terms (version 3.0) was performed via classical Fisher's exact test based on up-regulated and down-regulated terms, respectively. Multiple testing correction was performed using the Benjamini–Hochberg method. Data visualization was performed using the packages ggplot2 (v. 3.3.6) (Wickham, 2016) and pheatmap (v. 1.0.12) (Kolde, 2019).

### Western blot

Western blot assay was performed as previously described (Anelli et al, 2022). Briefly, cells were detached with Trypsin/0.5% EDTA (Thermo Fisher Scientific), counted with Neubauer chamber, and washed with ice-cold DPBS containing 10 mM N-ethylmaleimide (NEM) to block disulfide bond rearrangements; lysis was performed in 150 mM NaCl, 1% NP-40, 2% SDS, 50 mM Tris–HCl pH 7.4 containing 10 mM NEM and complete EDTA-free protease inhibitor cocktail (Roche) and phosphatase inhibitors (10 µl of lysis buffer for 100,000 cells). Samples were then processed with benzonase nuclease according to manufacturer's instructions, to remove nucleic acids. Lysate volumes corresponding to 50,000 cells were loaded under reducing conditions (50 mM DTT) on SDS–PAGE (4–20% gradient gel; Invitrogen). After transfer on nitrocellulose (0.2 µm; Ahmersham), membranes were incubated in blocking solution (5% wt/vol BSA, in TBS, 0.1% Tween 20) and incubated with primary and secondary antibodies prepared in TBS-Tween, 3% BSA. Signals were detected with Azure Sapphire FL Imager (Azure Biosystems).

Because of specific features of the target proteins, Western blot analysis of mitochondrial respiratory subunits was performed without thermal denaturation of the samples, and the transfer was performed on methanol-activated Immobilon-FL PVDF membrane (Millipore).

Quantification of signal intensity was performed by densitometry analysis with FiJi software. For each protein, signal was normalized on the control condition (D0 sample) and on the loading control (H3 histone).

### Semi-quantitative real-time PCR

Total RNA was extracted from T-HESC at different decidualization days by Trizol RNA lysis reagent (Invitrogen) and immediately stored at −80°C. When all the samples of the series were collected, they were retrotranscribed with MV reverse transcriptase (Promega) and random primers (Promega), according to manufacturer's instructions. Semi-quantitative real-time PCR (qRT-PCR) was performed with the iTAQ Universal SYBR Green Supermix (Bio-Rad) and the CFX96 Real-Time System (Bio-Rad).

Briefly, 0.2 µl of primer couples (25 µM each, to a final concentration of 0.5 µM), 5 µl of SYBR mix, and 4 µl of cDNA and

nuclease-free water (corresponding to 8 ng of template DNA) were added to each reaction well, in a total volume of 10 $\mu$l. All the samples were loaded in triplicate.

The amplification was performed with a denaturation step of 10 min at 95°C, followed by 40 cycles of PCR amplification (15 s at 95°C, 1 min at 60°C), and by a step-by-step temperature increase from 60°C to 95°C, to calculate the melting curve.

For each sample, the Ct (cycle number at which the fluorescence signal crosses the threshold) was measured, and the value of the replicates was averaged. To compare different conditions, the Ct of the reference gene GAPDH (whose expression is constant along the decidualization process) was subtracted from the Ct of the gene of interest in the same condition ($\Delta$Ct); then, each $\Delta$Ct was expressed in relation to the corresponding value in the control condition (by subtraction, giving the $\Delta\Delta$Ct), and expressed as fold change of gene expression (2-$\Delta\Delta$Ct).

Most primer sequences were chosen based on previous literature or on UCSC qRT-PCR primer database (Zeisel et al, 2013), as detailed below; primers for the KDELRs, instead, were purchased from QIAGEN. Primers with an optimal amplification temperature of 60°C were chosen.

### Statistical analyses

Statistical analyses were performed with GraphPad Prism 8.2. When required, previous calculations or data transformations were performed with Microsoft Excel. Parametric or non-parametric tests were performed as indicated in the figures, based on the features of the datasets analyzed. Statistical significance threshold was set at 0.05: $*P \le 0.05$; $**P \le 0.01$; $***P \le 0.001$; $****P \le 0.0001$; where asterisks are not indicated, $P > 0.05$. The statistic test used in each case is written in the corresponding figure legends.

## Data Availability

The datasets presented in this study can be found in online repositories. The names of the repository/repositories and accession number(s) can be found below: GEO accession: GSE200200.

## Supplementary Information

## Acknowledgements

Part of the present work was performed by M Dalla Torre in partial fulfillment of the requirements for obtaining the PhD degree at Vita-Salute San Raffaele University, Milano, Italy. The authors thank the ALEMBIC staff, in particular Valeria Berno, for assistance with immunofluorescence analysis and morphometry. We also thank Paola Panina-Bordignon and all members of our laboratories for fruitful discussions. The study was supported by research grants from AIRC (IG 2019 ID. 23285), and Ministero dell'Università e Ricerca (PRIN 2017XA5J5N) to R Sitia.

## Author Contributions

M Dalla Torre: conceptualization, data curation, software, formal analysis, validation, investigation, visualization, methodology, and writing—original draft, review, and editing.

D Pittari: data curation, software, formal analysis, and writing—review and editing.

A Boletta: conceptualization, methodology, and writing—review and editing.

L Cassina: data curation, investigation, methodology, and writing—original draft, review, and editing.

R Sitia: conceptualization, supervision, funding acquisition, and writing—original draft, review, and editing.

T Anelli: conceptualization, data curation, formal analysis, supervision, validation, investigation, project administration, and writing—original draft, review, and editing.

## Conflict of Interest Statement

The authors declare that they have no conflict of interest.

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
