## [Reviewer comments · Life Science Alliance]

Life Science Alliance

Mitochondria remodeling during endometrial stromal cell decidualization

Marco Dalla Torre, Daniele Pittari, Alessandra Boletta, Laura Cassina, Roberto Sitia, and Tiziana Anelli

DOI: <https://doi.org/10.26508/lsa.202402627>

Corresponding author(s): Tiziana Anelli, IRCCS Ospedale San Raffaele

Review Timeline:

Submission Date:	2024-01-29
Editorial Decision:	2024-03-22
Revision Received:	2024-09-17
Editorial Decision:	2024-09-18
Revision Received:	2024-09-24
Accepted:	2024-09-24

Transaction Report:

March 22, 2024

Re: Life Science Alliance manuscript #LSA-2024-02627-T

Dr Tiziana Anelli
San Raffaele Scientific Institute
DiBiT
via Olgettina 58
Milan, MI 20132
Italy

Dear Dr. Anelli,

Thank you for submitting your manuscript entitled "Mitochondria remodeling during endometrial stromal cell decidualization" to Life Science Alliance. The manuscript was assessed by expert reviewers, whose comments are appended to this letter. We invite you to submit a revised manuscript addressing the Reviewer comments.

Thank you for this interesting contribution to Life Science Alliance. We are looking forward to receiving your revised manuscript.

Sincerely,

B. MANUSCRIPT ORGANIZATION AND FORMATTING:

Reviewer #1 (Comments to the Authors (Required)):

The authors investigate decidualization of endometrial stromal cells (EnSCs) and how their differentiation affects their mitochondria and respiratory capacity. They show that mitochondrial network is increased and reorganized. They also show that mitochondria ER contacts are increased with decidualization. They additionally demonstrate that decidualized cells have increased mitochondrial respiration. Based on their previously published RNAseq dataset they also show that components of oxidative phosphorylation pathway are upregulated in decidualized cells, as are nuclear and mitochondrial encoded components. And they show that decidualization activates one of the master regulators of mitochondrial biogenesis.

Authors are discussing mitochondria ER contact in great detail in the manuscript, and they divide it into two sections. However, the data could be discussed in a single section. They indicate that this association likely reflects increased need to exchange of lipid biosynthesis intermediates but show no data reflecting that.

Additionally, their last point regarding master regulator activation could be expanded to include some of the other regulators, and show this activation by western blot by looking at nuclear fraction.

Sentence: "Western blotting analyses confirmed the increase of mitochondrial markers (Suppl. Figure 1A). The levels of TOM20 and VDAC (Voltage-Dependent Anion Channel), indeed, tended to progressively increase during" appears not to be finished.

Also, in Suppl. Figure 1A, authors use a portion of the Ponceau stained SDS-PAGE gel as a loading control for their quantification of protein levels. Since they show actin as well, have they quantified protein levels compared to actin as a loading control? Additionally, TOM20 and VDAC antibodies they use have multiple other bands show up, so I am unsure of the antibody specificity.

Typo "Most mitochondria had a normal structure and well-organized cristae. Coherently with what was observed in immunofluorescence..."

Could the authors include an explanation for MAM abbreviation in the figure legend for Figure 2B and C? I don't think abbreviation is explained anywhere in the text.

Typo in the sentence "Interestingly, the markers corresponding to components of the protein folding and quality control machinery of the ER (i.e. Protein Disulfide Isomerase, PDI, Figure 2C)..." are actually referring to Figure 2D.

Authors should include unit of time, hour if that is what it is, in Figure 4A and B.

In Figure 4B, authors show that transcripts involved in OXPHOS (mitochondrial) are high at the start of their experiment, decrease until 24 h timepoint, and then increase again. Can authors elaborate on this more since they mention in the text that both nuclear and the mitochondrial encoded components of OXPHOS are induced?

Could the authors give examples of transcripts they are referring to in the sentence "Genes related to mitochondrial replication, maintenance and translation in the mitochondria were instead downregulated... (Figure 4A-B)"? What about MFF and ATAD1 which are increased as shown in Figure 4B?

Just as a suggestion, could the authors include Supplementary table titles withing the file itself, above the tables/data they are showing? It would help reviewers while reading tables as the file names are not clearly indicated.

In Figure 5 authors show immunofluorescence analysis of PGC-1 α nuclear translocation. Have the authors tried to do western blot showing levels of nuclear fraction of PGC-1 α ? Showing this would strengthen their data. What do PGC-1 α levels look like during the entire differentiation? What about looking at some of the other regulators?

Authors indicate that EnSC switch to mitochondrial metabolism as development of spiral arteries provides the tissue with highly

oxygenated blood, however they talk about early events of decidualization, a switch from proliferative to secretory phase, which occurs earlier in pregnancy than the spiral artery remodeling. Unless their 6-day differentiation model is indicative of late pregnancy decidualized cell metabolism. Could they elaborate on that?

Figure 3D is slightly confusing for readers, it is showing that OCR/ECAR ratio is highly increased in D6 samples, and in the figure legend authors say that it quantifies changes in aerobic glycolysis. This leads the reader to assume that glycolysis is high. However, a higher OCR/ECAR ratio would indicate higher contribution of OXPHOS to their overall energy production. Authors need to explain this better in their legend and materials and methods.

Additionally, in the text authors mention "...and thus by a dramatic change in the OCR/ECAR ratio (Figure 3C)..." which is actually shown in Figure 3D. Is this a typo?

Authors show that mitochondria in decidualized EnSCs reorganize their mitochondria which elongate, however a publication by Ryu et al. in *Stem Cells and Development* from 2020 report that decidualized EnSCs have shorter mitochondria than those in proliferative phase. Ryu et al. also report that decidualized EnSCs have lower basal respiration and ATP production, which is opposite from findings reported by Authors of this manuscript. How do they explain these discrepancies?

Reviewer #2 (Comments to the Authors (Required)):

1. A short summary of the paper, including description of the advance offered to the field.

Torre et al. aims to characterize mitochondrial changes upon decidualization of endometrial stromal cells (EnSCs) and a potential mechanism as to how these changes occur. Their findings add on to a growing list of cell types where changes to mitochondrial fuel utilization and dynamics contribute to cellular differentiation.

2. For each main point of the paper, please indicate if the data are strongly supportive. If not, explicitly state the additional experiments essential to support the claims made and the timeframe that these would require.

- Upon decidualization, mitochondria elongate and increase mitochondria localization to cellular periphery.

1. How is decidualization confirmed? What are prominent cellular biomarkers? Cellular phenotypes?

2. How many cells were analyzed? Due to the limited sample size, does the significance hold if sample size was increased? Are statistics performed as if each cell is an individual N? Your text states 25 cells were assessed in 2 independent replicates. Are these experiments not an N=2 then? Cells utilized are an immortalized T-HESC cell line, hence obtaining cell number is not a limiting factor and at least 3-4 independent replicates should be performed.

3. How is mitochondrial expansion normalized to increased cellular volume? Is there a specific region of the cell where mitochondria are then defined as peripheral?

4. It is more convincing to show markers of decidualization to confirm mitochondrial cellular distribution is indeed due to differentiation.

5. Are there significances in SFig1A?

6. To fully convey an increase in mitochondrial mass, authors should consider probing for inner mitochondrial membrane protein, outer mitochondrial membrane protein, and matrix markers. The authors only assessed TOM20 levels, an outer mitochondrial membrane protein.

- Mitochondria have enhanced MERC (Mitochondrial-ER contact sites), bioenergetic capacity, and changes in substrate utilization to support differentiation.

1. SFig2B, are the statistics performed on N=50 because of 50 individual cells? How many independent replicates was this performed on? Based on a small number of cells, will the significance hold with a larger sample size?

2. Authors should include the quantification of RTN4 and PDI co-localization events. A single representative image is not substantial.

3. The authors attribute enhanced OCR as a result of increased metabolic activity. OCR was normalized to cell count. Given an increase in TOM20 and enhanced mitochondrial-ER interactions, authors should address the possibility that enhanced OCR is due to increased mitochondrial mass or increased mitochondrial calcium.

4. Authors observe a decrease in glycolysis in D6 cells as well as enhanced FAO genes (CPT1A and CPT1C), authors should consider performing additional experiments to characterize and support the potential finding that there is a change in substrate oxidative capacity upon decidualization.

5. RNA levels are a great overview; however, authors should consider looking into select enzymes and/or proteins to fully assess if their protein levels match what is observed by qPCR.

6. Authors have touched on enhanced MERC but also downregulation of calcium import and homeostasis genes, but then end their discussion in calcium involvement as "outside the scope of the study." This reviewer proposes the authors to look into calcium efflux/influx or calcium levels in their system before fully dismissing the subject.

7. Authors suggest "mitochondria are enlarged and made more efficient, but not actively replicated" - have authors looked into mitochondrial number? This reviewer would expect to see graphs depicting an increase in mitochondrial size and reduced mitochondrial number. Is there a reduction in mitophagy?

8. The reviewer suggests the authors should do a more targeted assessment of fatty acid oxidation. Enhanced CPT1A/C and reduced ECAR are both promising, but a direct assessment will solidify the author's findings.

9. No conclusive experiments were performed to fully address if calcium or lipids are involved despite enhanced MERC or reduction of ECAR yet the authors state "unlike other professional secretors, they invest in mitochondria as their primary energy

source, and shift from calcium to lipids as the main currency exchanged at MERCS."

- Changes to mitochondria after decidualization are a result of PGC1-alpha translocation and increased mitochondrial biogenesis.

1. Authors suggest "mitochondria are enlarged and made more efficient, but not actively replicated". Authors should also address how PGC1-alpha fits into this statement.

2. Can mitochondrial changes still occur with a reduction in PGC1-alpha? In a PGC1-alpha KD will mitochondria still change morphology, OCR, and substrate utilization under decidualization? If mitochondrial biogenesis is impaired can decidualization still occur?

3. Lastly, indicate any additional issues you feel should be addressed (text changes, data presentation, statistics etc.).

- Improving clarity of statements by being less vague. Examples include:

1. In the introduction, "secrete limited amounts of proteins"

2. Sentences describing specific figure panels should actually match the graph's data and overall message.

3. Authors will benefit by adding in citations or previous literature supporting the authors' observation of how their morphological changes under decidualization reflect efficient quality control. Authors should consider papers looking into cristae reorganization.

4. Fig4A, "Metabolism" and "Detoxification" are vague terms.

5. Discussion sentence #2. "Our results indicate that decidualization EnSCs achieve increased secretion by improving the efficiency rather than the size of the factory."

- Complete the sentence located in 1st results section, 4th paragraph, last sentence.

- Graphs that are missing significance, indicate if data is non-significant.

1. Fig 1D, change how significance is presented. The text largely states "mitochondria longer than 10um increase even more." Are all sizes significant or is the significance only for 10um mitochondria? A multiplier by how much the increase is a more valuable description than "increase even more."

- Data presentation:

1. Figure 1D, micron symbol is not complete.

2. Fig 2D, D0 images show multiple cells reflecting the phenotype described however D6 images are of only one cell. It is beneficial to use a representative image with multiple cells in one field of view.

3. Fig3A-C, authors should include in the y-axis how the graphs have been normalized. Instead of "A.U." authors should consider "cell count"

4. Fig 3, authors should indicate in the text/figure legend how many technical replicates were performed per individual experiment.

5. Is there value in showing both Fig 3C and Fig3D in the same panel? One of the panels can be supplemental.

Reviewer #1 (Comments to the Authors (Required))

The authors investigate decidualization of endometrial stromal cells (EnSCs) and how their differentiation affects their mitochondria and respiratory capacity. They show that mitochondrial network is increased and reorganized. They also show that mitochondria ER contacts are increased with decidualization. They additionally demonstrate that decidualized cells have increased mitochondrial respiration. Based on their previously published RNAseq dataset they also show that components of oxidative phosphorylation pathway are upregulated in decidualized cells, as are nuclear and mitochondrial encoded components. And they show that decidualization activates one of the master regulators of mitochondrial biogenesis.

Authors are discussing mitochondria ER contact in great detail in the manuscript, and they divide it into two sections. However, the data could be discussed in a single section. They indicate that this association likely reflects increased need to exchange of lipid biosynthesis intermediates but show no data reflecting that.

We thank the referee for the comment. The two sections regarding ER-mitochondria contact sites have been fused in one single section, as suggested. We have expanded the analysis of lipid metabolism by qPCR of numerous related genes (see page 7 and Suppl. Fig. 4 and 5). The data clearly indicate that decidualized EnSCs enhance this pathway, a notion that we now discuss in some depth (page 8).

Additionally, their last point regarding master regulator activation could be expanded to include some of the other regulators, and show this activation by western blot by looking at nuclear fraction.

Following the suggestion, we have performed WB analyses with antibodies against PGC1a. Unfortunately, the antibody yielded a plethora of background bands that confuse the results. To follow the referee's suggestion, we exploited qPCR analyses, now shown in revised Figure 5B, monitoring the mRNA levels of PGC-1alpha, NRF1 and NRF2.

Sentence: "Western blotting analyses confirmed the increase of mitochondrial markers (Suppl. Figure 1A). The levels of TOM20 and VDAC (Voltage-Dependent Anion Channel), indeed, tended to progressively increase during" appears not to be finished.

We thank the referee for noting that the word "decidualization" was missing. The mistake has been amended.

Also, in Suppl. Figure 1A, authors use a portion of the ponceau stained SDS-PAGE gel as a loading control for their quantification of protein levels. Since they show actin as well, have they quantified protein levels compared to actin as a loading control? Additionally, TOM20 and VDAC antibodies they use have multiple other bands show up, so I am unsure of the antibody specificity.

In the first version of the manuscript we had used actin and ponceau staining as a double control, normalizing densitometric analyses on total protein load (ponceau staining) as in our hands it is more reliable than actin quantification. The actin band is always very intense, even if not saturated. However, reasoning on the issues raised by the referee, we have decided to quantify the WB using another normalizer (Histone H3), as a count for cell number. We have also expanded our Western Blot analyses, following the behavior of different proteins for the different mitochondrial sub-regions (TOM20 and VDAC for the outer membrane, TIM23 for the inner membrane, Citrate synthase for the matrix) as well as different components of the OXPHOS complexes. A representative WB image (out of 3 performed) and the densitometric quantifications are presented in new Figure 1 C and D respectively. As to the TOM20 and VDAC antibodies, new better Western Blot panels have been inserted. We are sure of the identity of the bands identified because of the intense clear signal of the expected molecular weight.

Typo "Most mitochondria had a normal structure and well-organized cristae. Coherently with what was observed in immunofluorescence..."

The sentence is referring to the IF analysis shown in Figure 1, which is now cited in brackets to make the sentence more clear (page 5).

Could the authors include an explanation for MAM abbreviation in the figure legend for Figure 2B and C? I don't think abbreviation is explained anywhere in the text.

We have corrected the Figure substituting the acronym MAM with the one used in the text (MERC).

Typo in the sentence "Interestingly, the markers corresponding to components of the protein folding and quality control machinery of the ER (i.e. Protein Disulfide Isomerase, PDI, Figure 2C)..." are actually referring to Figure 2D.

We thank the referee: we have inserted the correct reference (Suppl. Figure 3 in the new revised version).

Authors should include unit of time, hour if that is what it is, in Figure 4A and B.

The mistake has been corrected; the unit of time is Hours.

In Figure 4B, authors show that transcripts involved in OXPHOS (mitochondrial) are high at the start of their experiment, decrease until 24 h timepoint, and then increase again. Can authors elaborate on this more since they mention in the text that both nuclear and the mitochondrial encoded components of OXPHOS are induced?

We thank the referee for this stimulating comment; we have inserted a sentence in the results sections ("Interestingly, mitochondrial-encoded subunits are downregulated initially but start to accumulate 36 hours after hormonal stimulation, possibly underlying a more complex regulation pattern. ").

Could the authors give examples of transcripts they are referring to in the sentence "Genes related to mitochondrial replication, maintenance and translation in the mitochondria were instead downregulated... (Figure 4A-B)"? What about MFF and ATAD1 which are increased as shown in Figure 4B?

Thanks to this comment, we have rephrased the sentence into "Finally, genes related to mitochondrial DNA replication, maintenance and repair are instead downregulated (Figure 4A-B), probably reflecting a switch from proliferation (in which the mitochondria are duplicated at every cell replication), to terminal decidualization."

The genes related to mt DNA replication, repair and maintenance, which from the transcriptomics data are downregulated at day 6 of decidualization are the following: LIG3, MGME1, DNA2, TOP3A, UNG, RECQL4, PRIMPOL, POLQ, METTL4, PIF1, ATAD3A.

The list at the different time points can be obtained from Suppl. material.

As to MFF (which is associated to mitochondrial dynamics) and ATAD1 (needed to remove mis-localized proteins from the mitochondrial outer membrane) we finally decided to exclude them to the heat-map and insert in the main figure only the above-mentioned genes.

Just as a suggestion, could the authors include Supplementary table titles within the file itself, above the tables/data they are showing? It would help reviewers while reading tables as the file names are not clearly indicated.

We apologize with the referee, it was a problem in uploading the data; we have solved it.

In Figure 5 authors show immunofluorescence analysis of PGC-1 α nuclear translocation. Have the authors tried to do western blot showing levels of nuclear fraction of PGC-1 α ? Showing this would strengthen their data. What do PGC-1 α levels look like during the entire differentiation? What about looking at some of the other regulators?

As discussed above, since our anti PGC1 α antibodies were unsatisfactory in WB, we analyzed its mRNA levels, and the levels of other regulators of mitochondria biogenesis. The data are now inserted in revised Figure 5B.

Authors indicate that EnSC switch to mitochondrial metabolism as development of spiral arteries provides the tissue with highly oxygenated blood, however they talk about early events of decidualization, a switch from proliferative to secretory phase, which occurs earlier in pregnancy than the spiral artery remodeling. Unless their 6-day differentiation model is indicative of late pregnancy decidualized cell metabolism. Could they elaborate on that?

Thank for this stimulating comment: differentiation entails a rapid mitochondrial "upgrading", detectable already a few hour after hormonal stimulation. These changes probably reflect a developmental program preparing these cells to operate (physiologically) near the spiral arteries. We have better discussed the issue in the text ("Yet, the metabolic

rewiring observed in our experiments is coherent with known aspects of physiological decidualization. Indeed, the development of spiral arteries provides the tissue with highly oxygenated blood (Pijnenborg 2006), which would justify the early mitochondrial network enlargement and final EnSC switch to increased oxygen consumption.”).

Figure 3D is slightly confusing for readers, it is showing that OCR/ECAR ratio is highly increased in D6 samples, and in the figure legend authors say that it quantifies changes in aerobic glycolysis. This leads the reader to assume that glycolysis is high. However, a higher OCR/ECAR ratio would indicate higher contribution of OXPHOS to their overall energy production. Authors need to explain this better in their legend and materials and methods. Additionally, in the text authors mention "...and thus by a dramatic change in the OCR/ECAR ratio (Figure 3C)..." which is actually shown in Figure 3D. Is this a typo?

We thank the referee for raising this issue. In the revised version, we inserted an OXPHOS-Glycolysis plot to better show the shift towards OXPHOS (revised figure 3C), and edited the text accordingly (page 6).

Authors show that mitochondria in decidualized EnSCs reorganize their mitochondria which elongate, however a publication by Ryu et al. in Stem Cells and Development from 2020 report that decidualized EnSCs have shorter mitochondria than those in proliferative phase. Ryu et al. also report that decidualized EnSCs have lower basal respiration and ATP production, which is opposite from findings reported by Authors of this manuscript. How do they explain these discrepancies?

We thank the referee bringing to our attention this manuscript, which we read with great interest. Being aware of the fact that different cellular models (cell origins, cell culture conditions) can yield different results, we contacted the authors, to explore the possibility of using their cellular systems to compare some of the findings with ours. Unluckily, the authors were unable to help us with this issue, due to permission restrictions of their institute. However, in a nice discussion with the authors, they recognized that different conditions could account for different observations, and they suggested that we could discuss the implications in the manuscript. Based on this, we have now inserted a sentence on this issue in the discussion section and cited the work accordingly. Thanks for this remark that led to an interesting insight.

Reviewer #2 (Comments to the Authors (Required))

1. A short summary of the paper, including description of the advance offered to the field. Torre et al. aims to characterize mitochondrial changes upon decidualization of endometrial stromal cells (EnSCs) and a potential mechanism as to how these changes occur. Their findings add on to a growing list of cell types where changes to mitochondrial fuel utilization and dynamics contribute to cellular differentiation.
2. For each main point of the paper, please indicate if the data are strongly supportive. If not, explicitly state the additional experiments essential to support the claims made and the timeframe that these would require.

- Upon decidualization, mitochondria elongate and increase mitochondria localization to cellular periphery.

1. How is decidualization confirmed? What are prominent cellular biomarkers? Cellular phenotypes?

Decidualization is confirmed not only by cellular morphology but also, for each experiment, by qPCR analyses of relevant decidualization markers (as in Anelli et al., 2021, Pittari, Dalla Torre et al. 2022 and in previous manuscripts in the field), now shown in Panel A of Supplementary Figure 1.

2. How many cells were analyzed? Due to the limited sample size, does the significance hold if sample size was increased? Are statistics performed as if each cell is an individual N? Your text states 25 cells were assessed in 2 independent replicates. Are these experiments not an N=2 then? Cells utilized are an immortalized T-HESC cell line, hence obtaining cell number is not a limiting factor and at least 3-4 independent replicates should be performed.

Statistical analyses were performed for each experiment. As immortalized T-HESC are quite large, one can hardly analyze more than 1 cell in each microscopy field, making image acquisition very time consuming. The same is for the image analysis, which follows the image acquisition at the microscope. Nevertheless, we performed one more experiment and added it in the overall analysis. We now show the graphs in the new panel B of revised Figure 1 using different colors for the samples of the three different experiments analyzed, to highlight the reproducibility of the data.

3. How is mitochondrial expansion normalized to increased cellular volume?

We calculated the cell volume with the Huygens software on deconvolved images, using the staining for HLA1 to define the cell membrane (new revised Figure 1, panel B, upper graph). In the same way, we calculated mitochondrial volume (new revised Figure 1, panel B, lower left graph) and expressed it as a % of the total cell volume (new revised Figure 1, panel B, lower right graph). We apologize as we now realize that no clear explanation of these steps, was given in the original version. Thus, we modified the text and figure to clarify the protocol. Moreover, the pipelines for morphometric analysis have been uploaded as Supplementary Material.

Is there a specific region of the cell where mitochondria are then defined as peripheral?

We defined mitochondria as “peripheral” depending on their localization near the plasmamembrane. Our immunofluorescence images show mitochondria extending just beneath the plasma membrane in decidualized cells with respect to control cells.

4. It is more convincing to show markers of decidualization to confirm mitochondrial cellular distribution is indeed due to differentiation.

Please, refer to answer to point 1.

5. Are there significances in SFig1A?

Where not indicated in the graphs, statistical significance is missing, as better explained now in the dedicated Materials and Methods paragraph. Western Blot analyses coming from different experiments are tricky to quantify in absolute terms. Easier is to compare the relative inductions/decreases within each experiment. Following the suggestions of both referees, we have inserted additional mitochondrial markers in panels C and D of revised Figure 1. The analyses have been performed on three different experiments, and the densitometric quantifications is shown in panel D. Statistical significance is indicated by asterisks. It is clear the increasing trend (although not statistically significant) for a few markers, which would very likely become significant should we perform more experiments, but we believe that this will be simply time and materials consuming and will not add much to the overall message of the manuscript.

6. To fully convey an increase in mitochondrial mass, authors should consider probing for inner mitochondrial membrane protein, outer mitochondrial membrane protein, and matrix markers. The authors only assessed TOM20 levels, an outer mitochondrial membrane protein.

We thank the referee for this suggestion, which we followed analyzing, by WB, TIM23 (as a marker of the inner mitochondrial membrane), Citrate Synthase (as a protein of the mitochondrial matrix) and components of the respiratory chain. The data and their densitometric quantifications are now inserted in new revised figure 1 (panels C and D . See also our answer to the previous point). Immunofluorescence analyses, instead, were precluded because of the high background of the antibodies.

- Mitochondria have enhanced MERC (Mitochondrial-ER contact sites), bioenergetic capacity, and changes in substrate utilization to support differentiation.

1. SFig2B, are the statistics performed on N=50 because of 50 individual cells? How many independent replicates was this performed on? Based on a small number of cells, will the significance hold with a larger sample size?

The analysis was based on 3 independent experiments, corresponding to 83 images for control cells and 79 for decidualized cells. Most images represent a portion of an individual cell, even if occasional resampling of different regions of the same cell is possible, and a few images included two adjacent cells. This sample size corresponds to a total of 378 mitochondria identified in D0 cells and 473 in D6 cells.

2. Authors should include the quantification of RTN4 and PDI co-localization events. A single representative image is not substantial.

Even though we do appreciate this comment, we believe the experiments needed to address it with sufficient confidence would involve an effort that brings us beyond the scope of this manuscript. Studying ER-mitochondrial membrane association is a challenging task. It can be performed with EM (as we did and shown in Figure 2A), while a proper analysis and quantification in immunofluorescence requires the use of proximity ligation assay or bimolecular fluorescence complementation. Indeed, what normal immunofluorescence (labelling with ER and mitochondria markers) can show is the proximity of the two signals, which cannot be analyzed further because of the intrinsic technique resolution (about 200 nm). We cannot thus apply correctly a co-localization analysis (e.g. Pearson coefficient) to PDI-mitochondria or RTN4-mitochondria immunofluorescence couples, as this is not a real co-localization, but rather a simple proximity. We believe, in any case, that the images that we had inserted are representative of what is described (correctly, and with a statistical analysis) by EM in panels D-F of revised Figure 2. Hence, we decided to keep the images of the original panel B of Figure 2, but move them to revised Supplementary Figure 2 panel D.

3. The authors attribute enhanced OCR as a result of increased metabolic activity. OCR was normalized to cell count. Given an increase in TOM20 and enhanced mitochondrial-ER interactions, authors should address the possibility that enhanced OCR is due to increased mitochondrial mass or increased mitochondrial calcium.

We thank the referee for raising this issue. Of course, we expected increased respiration as cells increase their mitochondrial mass. We have further discussed this issue. The point is that even when data are normalized, it is notable that the single cell is using more respiration than glycolysis when it differentiates.

4. Authors observe a decrease in glycolysis in D6 cells as well as enhanced FAO genes (CPT1A and CPT1C), authors should consider performing additional experiments to characterize and support the potential finding that there is a change in substrate oxidative capacity upon decidualization.

We thank the referee for this intriguing comment. We further investigated these metabolic changes by qPCR analyses of key genes in different metabolic pathways. New data are thus inserted in new revised Supplementary Figure 5 and discussed in the results section and in the discussion. We also evaluated the possibility of performing specific experiments to further characterize changes in energy sources in decidualizing cells. However, we reasoned that this would represent an entire new research project, for the experiments to be properly and satisfactorily performed.

5. RNA levels are a great overview; however, authors should consider looking into select enzymes and/or proteins to fully assess if their protein levels match what is observed by qPCR.

As the referees underlines, the mRNA levels are, of course, only a piece of information, as different processes can further modify the final amount on enzymes and proteins. As

suggested, we analyzed a few selected molecules by WB assays. A comprehensive proteomic analysis will be part of a next research project.

6. Authors have touched on enhanced MERC but also downregulation of calcium import and homeostasis genes, but then end their discussion in calcium involvement as "outside the scope of the study." This reviewer proposes the authors to look into calcium efflux/influx or calcium levels in their system before fully dismissing the subject.

Tickled by the intriguing questions of the referee, we analyzed more thoroughly the different MCU components (which is now part of new revised Supplementary Figure 4B). With our surprise, our data show that while MCUB decreases along decidualization, almost all the other components of the mitochondrial calcium uniporter show a first decrease and then a dramatic increase in the latest differentiation time points. A proper analysis of calcium efflux/influx is a most interesting development, which we hope to get funded. To be exhaustive, it would require the insertion of sensors in different compartments of the cells. We are indeed proceeding with a more comprehensive analysis of general "environmental, chemical" changes (redox, pH, zinc, calcium) in different compartments of the secretory pathway of decidualized endometrial stromal cells, which will be part of a future manuscript.

7. Authors suggest "mitochondria are enlarged and made more efficient, but not actively replicated" - have authors looked into mitochondrial number? This reviewer would expect to see graphs depicting an increase in mitochondrial size and reduced mitochondrial number. Is there a reduction in mitophagy?

The analysis performed on microscopy images was able to identify in total 2464 mitochondria in D0 images and 3321 in D6 images. On average, a cell at D0 contains 145 mitochondria, while a cell at D6 301. Hence, the mitochondria number does not look like decreasing, even if we are aware of the fact that we have used, for this analysis, a reduced number of cells and hence the mitochondria/cell number could be not representative of the entire population. Hence, we have not inserted the mitochondria/cell number in the manuscript. Of course mitophagy is a very interesting issue, that should however be studied with dedicated tools. Decidualized cells are designed to be long lived after embryo implantation which might entail stringent mitochondrial quality control. A sentence has been inserted in the results section (page 5).

8. The reviewer suggests the authors should do a more targeted assessment of fatty acid oxidation. Enhanced CPT1A/C and reduced ECAR are both promising, but a direct assessment will solidify the author's findings.

Please, refer to answer to point 4

9. No conclusive experiments were performed to fully address if calcium or lipids are involved despite enhanced MERC or reduction of ECAR yet the authors state "unlike other professional secretors, they invest in mitochondria as their primary energy source, and shift from calcium to lipids as the main currency exchanged at MERCS."

The referee is right in underlining this point. We have modified the discussion section, also according to the new results described in the answer to point 4 and inserted in the revised manuscript in Supplementary Figure 5B and in the Results and Discussion (page 7 and 8 respectively).

- Changes to mitochondria after decidualization are a result of PGC1-alpha translocation and increased mitochondrial biogenesis.

1. Authors suggest "mitochondria are enlarged and made more efficient, but not actively replicated". Authors should also address how PGC1-alpha fits into this statement.

2. Can mitochondrial changes still occur with a reduction in PGC1-alpha? In a PGC1-alpha KD will mitochondria still change morphology, OCR, and substrate utilization under decidualization? If mitochondrial biogenesis is impaired can decidualization still occur?

We thank the referee for raising this issue. We tried to answer to this question by WB analysis of PGC-1alpha translocation from the post-nuclear supernatants to the pellet after cell lysis in NP-40 buffer, but unfortunately, the antibody is not working satisfactorily in WB. We thus acquired new images of PGC-1alpha translocation in WB (to increase the numbers for statistical analysis) and measured by qPCR the levels of key transcription factors for mitochondrial biogenesis. We thus reported a steadily increase of PGC-1alpha from day 2 of decidualization, and an increase of Nrf2 at later time points; no increase instead can be seen for TFAM (not shown). We have inserted the data on PGC-1alpha, Nrf2 and Nrf1 in the revised version of Figure 5 (panel B) and discussed them in the results and in the discussion section. Whether decidualization can still occur without mitochondrial biogenesis is a difficult question: by inhibiting mitochondria biogenesis we would inhibit also physiological mitochondria renewal, crucial for cell survival. Indeed, mouse models KO for these factors show many defects in different organs, which makes difficult to get information about these factors and decidualization/fertility from murine model studies.

3. Lastly, indicate any additional issues you feel should be addressed (text changes, data presentation, statistics etc.).

- Improving clarity of statements by being less vague. Examples include:
 1. In the introduction, "secrete limited amounts of proteins"

The statement has been modified.

2. Sentences describing specific figure panels should actually match the graph's data and overall message.

We modified them accordingly.

3. Authors will benefit by adding in citations or previous literature supporting the authors' observation of how their morphological changes under decidualization reflect efficient quality control. Authors should consider papers looking into cristae reorganization.

We thank the reviewers for this suggestion. We have now inserted relevant references to the literature about the topic.

4. Fig4A, "Metabolism" and "Detoxification" are vague terms.

Metabolism and Detoxification are indeed very broad GO terms, that represent the first outputs of our Over Representation Analysis. Their enrichment is highly statistically significant, but the detail of the information they provide is very low. Accordingly, narrower "child terms", including fewer genes, are included in the analysis; given their features, the statistical power of the results is lower, but the level of detail provided is higher.

.5. Discussion sentence #2. "Our results indicate that decidualization EnSCs achieve increased secretion by improving the efficiency rather than the size of the factory."

As discussed in the following sentences in the discussion section, as well as explained in the introduction, while other differentiation programs to a secretory phenotype (e.g. B lymphocyte to plasma cell differentiation) require a dramatic enlargement of the secretory pathway, and in particular of the ER, decidualization entails a relative expansion of only the Golgi compartment, while the volume of the ER is instead decreasing if compared to the entire cell volume. Hence, the differentiation to a secretory phenotype is not achieved by a simple increase in the secretory compartment (= protein factory) but likely efficiency is improved.

- Complete the sentence located in 1st results section, 4th paragraph, last sentence.

Done.

- Graphs that are missing significance, indicate if data is non-significant.

A sentence explaining clearly that, in the absence of other indications, the data are NOT statistically significant has been inserted in the material and methods, in the paragraph regarding statistical analyses.

- Fig 1D, change how significance is presented. The text largely states "mitochondria longer than 10um increase even more." Are all sizes significant or is the significance only for 10um mitochondria? A multiplier by how much the increase is a more valuable description than "increase even more."

Done.

- Data presentation:

1. Figure 1D, micron symbol is not complete.

Changed.

2. Fig 2D, D0 images show multiple cells reflecting the phenotype described however D6 images are of only one cell. It is beneficial to use a representative image with multiple cells in one field of view.

This is why we choose images with multiple cells for day 0. At day 6 cells are so large that almost only one can fit in the high magnification field (see also answer to major point 2). We thus choose one representative image. The conclusion is supported by statistical analyses of numerous experiments.

3. Fig3A-C, authors should include in the y-axis how the graphs have been normalized. Instead of "A.U." authors should consider "cell count"

The y-axis (pmol/min/a.u.) already includes normalization, since OCR data are normalized in each well by the fluorescence signal (a.u.) of the CyQuant Cell Proliferation Assay (Thermo Fisher Scientific, #C35011), as explained in the method section.

4. Fig 3, authors should indicate in the text/figure legend how many technical replicates were performed per individual experiment.

Done.

5. Is there value in showing both Fig 3C and Fig3D in the same panel?

Panel C of the original Figure 3 was moved in Supplementary Figure 3 of the revised version, while panel D was substituted with an energy plot, showing basal OCR vs ECAR measurements.

September 18, 2024

RE: Life Science Alliance Manuscript #LSA-2024-02627-TR

Dr. Tiziana Anelli
IRCCS Ospedale San Raffaele
DiBiT
via Olgettina 58
Milan, MI 20132
Italy

Dear Dr. Anelli,

Thank you for submitting your revised manuscript entitled "Mitochondria remodeling during endometrial stromal cell decidualization". We would be happy to publish your paper in Life Science Alliance pending final revisions necessary to meet our formatting guidelines.

- please be sure that the authorship listing and order is correct
- please add a separate conflict of interest statement to your manuscript text
- please use the [10 author names, et al.] format in your references (i.e. limit the author names to the first 10)
- please add figure callouts for Figure 5 A,C to your main manuscript text
- the accession information for the RNA-seq data should be mentioned in a separate Data Availability statement placed at the end of the Materials and Methods section

Figure Check:

- For Figure 5, you have a panel C in your figure legend, but this does not appear in the figure. Please correct.
- For Figure S2, you have panels A-D in the figure, but there are only panels A,B in the figure legend. Please correct.

LSA now encourages authors to provide a 30-60 second video where the study is briefly explained. We will use these videos on social media to promote the published paper and the presenting author (for examples, see <https://docs.google.com/document/d/1-UWCfbE4pGcDdcgzcmiuJl2XMBJnxKYeqRvLLrLS08s/edit?usp=sharing>). Corresponding or first-authors are welcome to submit the video. Please submit only one video per manuscript. The video can be emailed to contact@life-science-alliance.org

A. FINAL FILES:

B. MANUSCRIPT ORGANIZATION AND FORMATTING:

Sincerely,

September 24, 2024

RE: Life Science Alliance Manuscript #LSA-2024-02627-TRR

Dr. Tiziana Anelli
IRCCS Ospedale San Raffaele
DiBiT
via Olgettina 58
Milan, MI 20132
Italy

Dear Dr. Anelli,

Thank you for submitting your Research Article entitled "Mitochondria remodeling during endometrial stromal cell decidualization". It is a pleasure to let you know that your manuscript is now accepted for publication in Life Science Alliance. Congratulations on this interesting work.

DISTRIBUTION OF MATERIALS:

Again, congratulations on a very nice paper. I hope you found the review process to be constructive and are pleased with how the manuscript was handled editorially. We look forward to future exciting submissions from your lab.

Sincerely,
